# WHAT YOU THINK IS WHAT YOU SEE:
# Driving Exploration in VLM Agents via Visual-Linguistic Curiosity

**Haoxi Li** [1]  **Qinglin Hou** [2]  **Jianfei Ma** [3]  **Jinxiang Lai** [1]  **Tao Han** [1]  **Sikai Bai** [1]  **Jingcai Guo** [4]
**Jie Zhang** [1]  **Song Guo** [1]

## Abstract

To navigate partially observable visual environments, recent VLM agents increasingly internalize world modeling capabilities into their policies via explicit CoT reasoning, enabling them to mentally simulate futures before acting. However, relying solely on passive reasoning over visited states is insufficient for sparse-reward tasks, as it lacks the epistemic drive to actively uncover the *"known unknown"* required for robust generalization. We ask: *Can VLM agents actively find signals that challenge and refine their internal world model through curiosity-driven exploration?* In this work, we propose GLANCE, a unified framework that bridges reasoning and exploration by grounding the agent's linguistic world model into the stable visual representations of an evolving target network. Crucially, GLANCE leverages the discrepancy between linguistic prediction and visual reality as an intrinsic curiosity signal within reinforcement learning, steering the agent to actively explore areas where its internal model is uncertain. Extensive experiments across a series of agentic tasks show the effectiveness of GLANCE, and demonstrate that aligning "what the agent thinks" with "what the agent sees" is key to solving complex or sparse agentic tasks.

## 1. Introduction

The rapid evolution of Vision-Language Model (VLM) agents has fundamentally shifted the control paradigm from

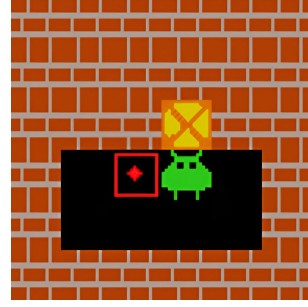 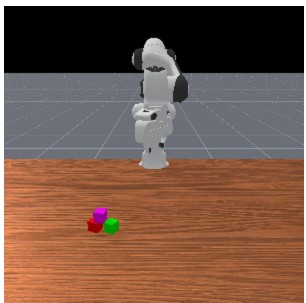

*(a)* Sokoban Observation    *(b)* PrimitiveSkill Observation

*Figure 1.* **Prediction error** from different environments. (a) *Prediction:* the target and box will be at the same row. (b) *Prediction:* the purple cube will be stacked on top of the green cube.

text-based, fully observable settings to complex, partially observable visual environments. To navigate these complex dynamics, recent VLM agents increasingly internalize world modeling (Xing et al., 2025) directly into their policies via reinforcement learning (RL), employing explicit Chain-of-Thought (CoT) reasoning to maintain belief states and predict world dynamics. However, current paradigms (Wang et al., 2025; Chen et al., 2025; Shu et al., 2025; He et al., 2025) largely restrict this capability to *passive exploitation* of visited states, refining the agent's reasoning accuracy on the current data distribution rather than driving active discovery. This reliance on passive interpretation often leads to spurious success in sparse-reward settings: an agent may learn to perfectly describe a *"dead end"* in a puzzle game without ever realizing it should have explored a different path. We thus ask: *Can VLM agents actively find signals that challenge and refine their internal world model through curiosity-driven exploration?*

To address this, *curiosity learning* (Schmidhuber, 1991; Sun et al., 2011; Schmidhuber, 2010; Houthooft et al., 2016; Azar et al., 2019) offers a natural solution. Typically, it involves training an auxiliary world model to predict future observations, using the prediction error as an intrinsic reward. However, applying this paradigm directly to VLM agents overlooks a critical architectural shift: the world model is no longer an external auxiliary module but is internalized within the policy itself via linguistic CoT reasoning. Consequently, standard curiosity methods (Pathak et al.,

---

[1]Department of Computer Science and Engineering, Hong Kong University of Science and Technology, Hong Kong, China. [2]Department of Computer Science, University of Southern California, Los Angeles, USA [3]School of Computing, National University of Singapore, Singapore [4]Department of Computing, Hong Kong Polytechnic University, Hong Kong, China. Correspondence to: Jie Zhang <csejzhang@ust.hk>.

*Proceedings of the 43rd International Conference on Machine Learning*, Seoul, South Korea. PMLR 306, 2026. Copyright 2026 by the author(s).

2017; Ermolov & Sebe, 2020; Groth et al., 2021; Guo et al., 2022), which focus solely on predicting latent visual futures from visual pasts, are insufficient, as they decouple exploration from the agent's linguistic reasoning process. For instance, a VLM agent might learn robust visual representations to satisfy the curiosity objective, yet its linguistic reasoning could remain detached from physical reality, leading to hallucinations. To bridge this modality gap, we argue for a unified objective grounded in the principle that *"what the VLM agent thinks should predict what it sees"*. Effective exploration, therefore, must drive the VLM agent towards interactions where its linguistic hypothesis of the future fails to align with the visual reality, forcing the internalized world model to ground itself through active falsification.

In this work, we present GLANCE (**G**rounding **L**inguistic **A**lig**n**ment for **C**uriosity **E**xploration), a unified framework that operationalizes this principle by strictly aligning the agent's explicit reasoning with visual reality. Specifically, GLANCE projects the latent representation of the VLM agent's *linguistic prediction* into the *visual representation space* of a momentum-updated target network, which encodes the actual observation. We leverage the discrepancy between these two representations as a unified self-supervised objective that solves three problems simultaneously: (i) shapes the visual encoder to capture semantically actionable features, (ii) grounds the internalized world model in physical dynamics, and (iii) provides a curiosity-driven intrinsic reward to train the policy. This joint optimization of world modeling and exploration transforms the latter from stochastic search into active falsification, steering the agent towards states where its current reasoning fails to explain the visual reality. Unlike standard visual bootstrapping methods (Grill et al., 2020; Schwarzer et al., 2020; Guo et al., 2022) that rely solely on visual-to-visual prediction, GLANCE drives world modeling via natural language reasoning, ensuring exploration is guided by the agent's semantic understanding rather than mere visual novelty.

However, a critical challenge arises from the mismatch in learning dynamics between the VLM agents' *thinking* and *seeing*. Specifically, since the pre-trained LLM backbone is already semantically rich, the proposed lightweight alignment projector tends to quickly overfit to superficial visual features. This leads to an early vanishing in intrinsic rewards, a phenomenon we term *curiosity drain*, where the agent stops exploring because it mistakenly believes it has mastered the environment. To address this, we further introduce a *Curriculum Exploration* mechanism. By periodically re-initializing the projector weights while preserving the evolving visual encoder, we enforce a successive refinement process. This rejuvenation compels the agent to re-examine familiar states using its enhanced visual perception, uncovering fine-grained discrepancies that were previously masked by the projector's overfitting. Conse-

quently, exploration becomes a self-paced curriculum: as the visual encoder captures increasingly complex semantics, the rejuvenated projector continuously reveals new layers of *known unknowns*, sustaining the curiosity drive throughout long-horizon learning.

We empirically evaluate GLANCE across a diverse suite of four agentic task domains: Grid Puzzles, Navigation, Object Manipulation, and Geometric Reconstruction. Our results demonstrate that GLANCE consistently outperforms current exploitation-based RL methods (Wang et al., 2025; Chen et al., 2025) in VLM agents. Notably, we show that our cross-modal curiosity empowers VLM agents to learn effective exploration policies even in the absence of extrinsic rewards. Furthermore, ablation studies on *Curriculum Exploration* confirm that periodic rejuvenation of the alignment projector is crucial to preventing curiosity collapse and sustaining long-term epistemic drive. Remarkably, GLANCE achieves these results using a lightweight architecture where the VLM agent simultaneously serves as the world model and the policy, concurrently trained across all tasks without human demonstrations. These findings suggest that the synergetic grounding of *thinking* and *seeing* is a fundamental prerequisite for building autonomous, curious, and physically-grounded agents.

## 2. Preliminaries

In this section, we formalize the reinforcement learning framework for VLM agents and describe the internalized world modeling paradigm that serves as our foundation.

### 2.1. Problem Formulation

We formulate the multi-turn VLM agentic task as a Partially Observable Markov Decision Process (POMDP) (Åström, 1965; Kaelbling et al., 1998), defined by a tuple $(\mathcal{S}, \mathcal{A}, \mathcal{O}, T, R, Z, \gamma)$, where $\mathcal{S}$, $\mathcal{A}$, and $\mathcal{O}$ correspond to the state, action, and observation spaces. At each turn $t$, the agent takes an action $a_t \sim \pi_{\boldsymbol{\theta}}(a_t|b_t)$ in a sequence of natural language tokens, where $b_t$ is a sufficient statistic computed from the history $h_t = (o_0, a_0, \ldots, o_{t-1}, a_{t-1}, o_t)$. Then, the environment transitions to a new state $s_{t+1} \sim T(s_{t+1}|s_t, a_t)$, which is hidden from the agent's view, and emits a visual observation $o_{t+1} \sim Z(o_{t+1}|s_{t+1}, a_t)$ and a reward $R(s_t, a_t)$ to the agent. For brevity, we will refer to the reward using the shorthand $r_t$ whenever applicable. The VLM agent's objective is to learn a policy $\pi_{\boldsymbol{\theta}}$ that maximizes the expected cumulative discounted return:

$$\mathcal{J}(\boldsymbol{\theta}) = \mathbb{E}_{\pi_{\boldsymbol{\theta}}, T, Z} \left[ \sum_{t=0}^{\infty} \gamma^t \mathbb{E}_{b_t}[R(s_t, a_t)] \,\middle|\, b_0 \right] \quad (1)$$

where $\gamma \in [0, 1)$ is the discount factor, $b_0$ is the initial belief, and $b_t$ is updated deterministically as $b_t =$

$\tau(b_{t-1}, a_{t-1}, o_t)$, where $\tau$ is the belief update function.

Since solving the belief-space POMDP is computationally intractable (Madani et al., 1999), history-based methods (Silver & Veness, 2010; Guez et al., 2012) provide a tractable alternative by summarizing past observations. Analogously, RNN- or Transformer-based architectures can be seen as maintaining a hidden state that encodes this history, and, in this view, a VLM can be interpreted as updating its internal representation with each new observation to predict the next token, akin to approximate belief tracking in a POMDP.

## 2.2. Internal World Models in VLM Agents

**Reasoning as World Modeling** To address the demands of partial observability, recent frameworks (Wang et al., 2025) require VLMs to explicitly internalize world modeling into their reasoning processes. Specifically, at turn $t$, upon receiving an observation $o_t$, the model infers the underlying state $s_t$ that generates it. From this inferred state, a reasoning path $z_t$ is constructed, and a prediction of the next state transition $s_{t+1}$ is made, which serves as a prior before the next observation is received. Together, the inferred state, reasoning path, and predicted next state constitute the agent's internal world-modeling tokens $\Phi$, structured as the following sequence:

`<Obs>`$s_t$`</Obs><Res>`$z_t$`</Res><Pred>`$s_{t+1}$`</Pred>`.

By combining the inferred state $s_t$ and the reasoning trace $z_t$, the model produces an action $a_t$ (`<Ans>`$a_t$`</Ans>`) that is executable in the environment. Notably, the predicted next state reflects the agent's belief about future outcomes. The knowledge gap is often characterized by the discrepancy between this prediction and the actual observation, highlighting potential weaknesses in the agent's world modeling capability.

**Optimization with Extrinsic Rewards** Following the existing VLM-RL paradigm, the policy is typically optimized using the Proximal Policy Optimization (PPO) algorithm (Schulman et al., 2017). Such VLM agent is supervised by a composite reward signal provided by the environment or external evaluators (e.g., LLM-as-a-judge). We denote this aggregate signal as the *extrinsic reward* $r_t^e$:

$$r_t^e = r_t^{\text{task}} + r_t^{\text{reason}} + r_t^{\text{format}} \qquad (2)$$

where $r_t^{\text{task}}$ is the sparse task reward, $r_t^{\text{reason}}$ evaluates the quality of world model reasoning, and $r_t^{\text{format}}$ ensures structural adherence. To handle the hierarchical credit assignment between turn-level outcomes and token-level generation, a Bi-Level Generalized Advantage Estimation (GAE) mechanism is employed, which propagates advantages from the final turn-level reward back to individual reasoning tokens within each action sequence.

## 3. Method

In this section, we propose GLANCE, a unified framework designed to transform the VLM agent's internalized world modeling into a structured source of epistemic drive. As illustrated in Fig. 2, the architecture consists of two parallel streams: an *Online VLM Agent* that generates reasoning and visual-grounded predictions, and a *Target Network* that provides stable evolving regression targets.

### 3.1. The GLANCE Framework

The core design of GLANCE is built upon the interaction between a thinking policy and a witnessing encoder. We formalize the two primary components as follows.

**Online VLM Agent** The online stream is parameterized by $\theta = (\mathbf{v}, \ell)$, comprising a visual encoder $f_\mathbf{v}$ and a LLM backbone $\Lambda_\ell$. At each turn $t$, given the history $h_t$ and the current observation $o_t$, the agent internalizes the world modeling with $\Phi_t$, followed by an action $a_t$. To account for the prediction error, we define the *linguistic hypothesis state* $h_{t+1} \in \mathbb{R}^d$ as the Transformer's final layer hidden state corresponding to the position of the last prediction token. This vector $h_{t+1}$ serves as a semantic compression of the agent's prediction about the future state $s_{t+1}$. To bridge the modality gap between the linguistic latent space and the visual feature space, we augment the online agent with a lightweight projector $g_\psi$.

**Momentum Encoder** To provide stable targets for the self-supervised alignment task, we instantiate a momentum target network parameterized by $\phi$. This stream consists of a visual encoder $f_\phi$, which is structurally identical to the online visual encoder $f_\mathbf{v}$. Following the principle of bootstrap latent methods (Grill et al., 2020; Guo et al., 2022), the parameters $\phi$ are not updated via gradient descent. Instead, the target network acts as a slowly evolving teacher that maintains a moving average of the online encoder's weights. At the end of each training iteration, the parameters are updated as:

$$\phi \leftarrow \alpha\phi + (1 - \alpha)\mathbf{v}, \qquad (3)$$

where $\alpha \in [0, 1]$ is the target decay rate. Upon executing the executable action $a_t^e$ and observing the next turn $o_{t+1}$, the target network encodes the future observation into a *visual reality representation*:

$$y_{t+1} = \text{sg}(f_\phi(o_{t+1})), \qquad (4)$$

where $\text{sg}(\cdot)$ denotes the stop-gradient operator. This setup enables GLANCE to align the agent's current linguistic hypothesis $h_{t+1}$ with the grounded visual reality $y_{t+1}$ in a temporally consistent manner.

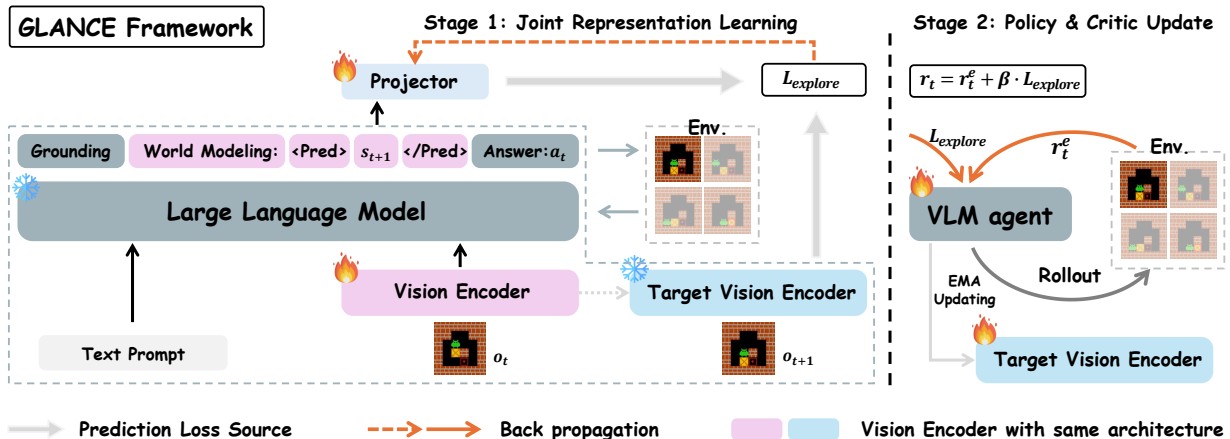

*Figure 2.* **Overview of the GLANCE framework.** GLANCE unifies world modeling and exploration into a self-supervised cross-modal loop. **Left:** The VLM agent generates an explicit reasoning trajectory containing a future state prediction $s_{t+1}$. The latent representation corresponding to the final `</pred>` token is mapped via a lightweight projector to align with the visual reality encoded by a Momentum Target Vision Encoder. The resulting prediction loss, $\mathcal{L}_{\text{explore}}$, serves as the source of curiosity. Crucially, gradients from $\mathcal{L}_{\text{explore}}$ back-propagate through the frozen LLM to update the Online Vision Encoder, forcing it to capture semantically actionable features. **Right:** The VLM agent interacts with the environment via rollouts, guided by a composite reward $r_t = r_t^e + \beta \cdot \mathcal{L}_{\text{explore}}$. This incentivizes the agent to proactively visit states where its internal reasoning fails to explain visual outcomes. The target network is updated via EMA to provide stable regression anchors. Fire icons indicate trainable modules, while snowflake icons denote frozen parameters.

### 3.2. Grounding Reasoning via Visual Alignment

To endow the agent with a physically grounded world model, we require that its *linguistic hypothesis* of the future dynamics (manifested in the reasoning tokens) effectively predicts the *visual reality* (encoded by the target network). We operationalize this principle through a self-supervised cross-modal prediction objective.

**Cross-modal Prediction**   Given the extracted linguistic hypothesis state $h_{t+1}$ from the online VLM's last prediction token, we first map it from the semantic language space to the visual latent space using the projector $g_\psi$. Formally, the online prediction $\widehat{y}_{t+1}$ is computed as:

$$\widehat{y}_{t+1} = g_\psi(h_{t+1}). \tag{5}$$

Simultaneously, the momentum target network processes the next-turn visual observation $o_{t+1}$ to yield the target representation $y_{t+1} = f_\phi(o_{t+1})$.

**Prediction Objective**   The prediction loss $\mathcal{L}_{\text{explore}}$ to minimize is defined as the mean squared error between the normalized predictions and targets:

$$\mathcal{L}_{\text{explore}}(\mathbf{v}, \boldsymbol{\psi}, t) = \left\| \frac{\widehat{y}_{t+1}}{\|\widehat{y}_{t+1}\|_2} - \text{sg}\left( \frac{y_{t+1}}{\|y_{t+1}\|_2} \right) \right\|_2^2, \tag{6}$$

where $\text{sg}(\cdot)$ denotes the stop-gradient operator. This operator is crucial for preventing representational collapse,

ensuring that the target network provides a stable anchor for the online agent to regress towards.

To ensure that the internalized world model is effectively grounded while maintaining the LLM's semantic priors, we optimize Eq. (6) through a selective gradient routing: we *freeze* the LLM parameters to prevent computational overhead and language drift (Lazaridou & Baroni, 2020), but allow gradients to back-propagate through the backbone. This procedure ensures that $\nabla \mathcal{L}_{\text{explore}}$ explicitly updates the projector $g_\psi$ and, crucially, the online visual encoder $f_\mathbf{v}$. By doing so, the visual encoder is shaped to extract features that are semantically consistent with the agent's logical reasoning, effectively *"learning to see what the agent thinks"*.

### 3.3. Curiosity as Active Exploration

To drive *active exploration*, we interpret the prediction error not merely as a representational loss, but as a proxy for epistemic uncertainty. The intrinsic reward $r_t^{\text{i}}$ at turn $t$ is formulated as follows:

$$r_t^{\text{i}} = \beta \cdot \mathcal{L}_{\text{explore}}(\mathbf{v}, \boldsymbol{\psi}, t). \tag{7}$$

The total reward signal $r_t$ utilized for policy optimization is defined as a composite sum:

$$r_t = r_t^{\text{e}} + r_t^{\text{i}}, \tag{8}$$

where $r_t^{\text{e}}$ is the extrinsic reward defined in Sec. 2.2.

*Table 1.* Main results on the general agentic benchmarks. We report the average success rate for puzzle and embodied control tasks, and perceptual similarity (Average of DINO and DreamSim) for the SVG reconstruction task. Trained models utilize QWEN2.5-VL-3B as the backbone. Bold indicates the best performance among trained models. See Appendix B.5 for multi-seed results.

| Model/Method | Sokoban | FrozenLake | Navigation | | | PrimitiveSkill | | | | | SVG | | | Overall |
|---|---|---|---|---|---|---|---|---|---|---|---|---|---|---|
| | | | Base | Common | **Average** | Place | Stack | Drawer | Align | **Average** | Dino | DreamSim | **Average** | |
| *Open-Source Models* | | | | | | | | | | | | | | |
| Qwen2.5-VL-72B (Bai et al., 2023) | 0.20 | 0.44 | 0.70 | 0.77 | 0.74 | **1.00** | 0.50 | 0.00 | **1.00** | 0.63 | 0.84 | 0.62 | 0.73 | 0.55 |
| Qwen2.5-VL-7B (Bai et al., 2023) | 0.14 | 0.14 | 0.33 | 0.38 | 0.35 | 0.00 | 0.00 | 0.00 | 0.75 | 0.19 | 0.84 | 0.27 | 0.56 | 0.28 |
| Qwen2.5-VL-3B (Bai et al., 2023) | 0.13 | 0.14 | 0.20 | 0.26 | 0.23 | 0.00 | 0.00 | 0.00 | 0.00 | 0.00 | 0.79 | 0.30 | 0.54 | 0.21 |
| VLM-R1-3B (Shen et al., 2025) | 0.16 | 0.15 | 0.33 | 0.34 | 0.34 | 0.00 | 0.00 | 0.00 | 0.00 | 0.00 | 0.79 | 0.27 | 0.54 | 0.24 |
| *Dense Extrinsic Rewards RL with World Model Reasoning for Visual States (Backbone: Qwen2.5-VL-3B)* | | | | | | | | | | | | | | |
| VAGEN-Full | 0.79 | 0.72 | 0.80 | 0.81 | 0.81 | **1.00** | **0.88** | **1.00** | **1.00** | 0.97 | 0.90 | 0.66 | 0.78 | 0.81 |
| GLANCE-Full | **0.85** | 0.78 | 0.86 | **0.88** | **0.87** | **1.00** | **0.88** | **1.00** | **1.00** | 0.97 | 0.92 | 0.70 | 0.81 | **0.86** |
| *Sparse Extrinsic Reward RL with World Model Reasoning Strategy (Backbone: Qwen2.5-VL-3B)* | | | | | | | | | | | | | | |
| VAGEN-Base | 0.61 | 0.71 | 0.78 | 0.80 | 0.79 | **1.00** | **0.88** | 0.88 | 0.88 | 0.91 | 0.90 | 0.65 | 0.78 | 0.76 |
| GLANCE-Base | 0.74 | 0.73 | 0.81 | 0.81 | 0.81 | **1.00** | **0.88** | **1.00** | 0.88 | 0.94 | 0.92 | 0.69 | 0.80 | 0.80 |
| *Turn-level PPO with World Model Reasoning Strategy (Backbone: Qwen2.5-VL-3B)* | | | | | | | | | | | | | | |
| Turn-PPO w/ Mask | 0.38 | 0.68 | 0.78 | 0.84 | 0.81 | 0.00 | 0.00 | 0.00 | **1.00** | 0.25 | 0.89 | 0.64 | 0.77 | 0.58 |
| GLANCE w/ Turn-PPO | 0.52 | 0.70 | 0.78 | 0.80 | 0.79 | **1.00** | 0.63 | 0.00 | **1.00** | 0.66 | 0.90 | 0.66 | 0.78 | 0.69 |
| *RL Baselines with World Model Reasoning Strategy (Backbone: Qwen2.5-VL-3B)* | | | | | | | | | | | | | | |
| Vanilla-PPO | 0.18 | 0.21 | 0.32 | 0.25 | 0.29 | 0.00 | 0.00 | 0.00 | 0.00 | 0.00 | 0.83 | 0.44 | 0.64 | 0.26 |
| GRPO w/ Mask | 0.20 | 0.57 | **0.88** | 0.81 | 0.85 | 0.00 | 0.00 | 0.00 | **1.00** | 0.25 | 0.92 | 0.66 | 0.79 | 0.54 |
| *Proprietary Models* | | | | | | | | | | | | | | |
| o4-mini (OpenAI, 2025) | 0.44 | **0.82** | 0.75 | 0.75 | 0.75 | **1.00** | 0.50 | 0.00 | 0.75 | 0.56 | 0.90 | 0.66 | 0.78 | 0.67 |
| GPT-4o (Hurst et al., 2024) | 0.43 | 0.54 | 0.75 | 0.69 | 0.72 | 0.50 | 0.63 | 0.00 | 0.88 | 0.50 | 0.91 | 0.69 | 0.80 | 0.60 |
| Gemini 2.5 Pro (Google, 2025) | 0.58 | 0.78 | 0.63 | 0.63 | 0.63 | 0.63 | 0.63 | 0.00 | 0.75 | 0.50 | 0.93 | 0.78 | 0.86 | 0.67 |
| Claude 4.5 Sonnet (Anthropic, 2025) | 0.31 | 0.80 | 0.67 | 0.67 | 0.67 | 0.63 | 0.50 | 0.00 | **1.00** | 0.53 | 0.95 | **0.81** | **0.88** | 0.64 |
| Claude 3.7 Sonnet (Anthropic, 2024) | 0.25 | 0.69 | 0.48 | 0.47 | 0.47 | 0.63 | 0.13 | 0.00 | **1.00** | 0.44 | 0.94 | 0.77 | 0.85 | 0.54 |

## 3.4. Curriculum Exploration with Rejuvenation

In GLANCE, a critical challenge arises from the mismatch in learning dynamics: the lightweight projector $g_\psi$ tends to converge rapidly to the pre-trained LLM's rich semantic space, finding trivial mappings to the visual targets long before the visual encoder $f_\mathbf{v}$ has fully adapted. This leads to an early vanishing of $\mathcal{L}_{\texttt{explore}}$, a phenomenon we term curiosity drain. To counteract this, we introduce an adaptive *Curriculum Exploration* mechanism.

Formally, we monitor the instantaneous rate of change in prediction loss, defined as

$$\delta_t = \big|\mathcal{L}_{\texttt{explore}}^{(t)} - \mathcal{L}_{\texttt{explore}}^{(t-1)}\big|. \qquad (9)$$

When $\delta_t$ remains below a convergence threshold $\epsilon$ for a consecutive duration of $K$ steps, we randomly re-initialize the projector parameters, while *preserving* the evolved parameters of the online visual encoder $\mathbf{v}$. Upon rejuvenation, the prediction error spikes. This compels the agent to revisit familiar states to ground its reasoning in increasingly fine-grained visual details, thereby sustaining the epistemic drive and revealing new layers of "known unknowns" throughout the learning process.

## 4. Experiment

In this section, we evaluate GLANCE against state-of-the-art VLM-RL baselines across five diverse agentic tasks. We aim to answer: (i) Can GLANCE facilitate effective exploration in sparse-reward environments? (ii) Does reasoning-driven curiosity improve the grounding of internalized world models? and (iii) How does the adaptive curriculum rejuvenation sustain long-term exploration?

### 4.1. Experimental Setup

**Environment** We benchmark GLANCE on a comprehensive evaluation suite featuring five distinct tasks that span 2D grid puzzles, 3D embodied navigation, and open-ended generative reasoning.

*Cognitive Grid Puzzles (Sokoban & FrozenLake):* These classic reasoning domains serve as testbeds for multi-step logical planning. In Sokoban (Schrader, 2018), the agent must execute a precise sequence of pushes to move boxes to targets while avoiding irreversible deadlocks. Frozen-Lake (Towers et al., 2024) requires navigating a hazardous grid where safe paths are non-trivial to discover. Both tasks are fully observable with discrete actions, while agents are

trained under the same history-based policy.

*Embodied 3D Control (Navigation & PrimitiveSkill):* To probe performance in high-dimensional visual settings, we employ Navigation (Yang et al., 2025; Kolve et al., 2017) and PrimitiveSkill (Tao et al., 2024; Nasiriany et al., 2022). Navigation is a first-person 3D task that requires the agent to locate specific objects in indoor scenes, challenging its ability to synthesize spatial memory from transient visual observations. PrimitiveSkill involves controlling a Panda robotic arm for complex manipulation using a hybrid action space (e.g., `pick(x,y,z)`). This task requires fine-grained visual grounding to map 3D coordinate reasoning to physical interactions.

*Generative Reasoning (SVG Reconstruction):* Representing an open-ended creativity task (Rodriguez et al., 2025), the agent must generate SVG code to replicate a target geometry. This task features an unbounded text-based action space and requires the agent to align abstract linguistic commands with dense, pixel-level visual outcomes, serving as a stress test for cross-modal consistency.

**Baselines** We utilize `Qwen2.5-VL-3B` (Bai et al., 2023) as the unified VLM backbone. We evaluate GLANCE as a plug-and-play module across three RL baselines with world model reasoning strategy to demonstrate its robustness: (i) VAGEN (Wang et al., 2025), featuring extrinsic rewards $r_t^e$ and Bi-Level GAE; (ii) VAGEN-Base (Wang et al., 2025), which utilizes more sparse rewards $r_t^{\text{format}} + r_t^{\text{task}}$ and standard token-level GAE; and (iii) Turn-level PPO with masking (Zhai et al., 2024). These are also compared against standard RL baselines including GRPO (Shao et al., 2024) and Vanilla PPO (Schulman et al., 2017). Furthermore, we compare our results against zero-shot performance from seven existing large vision-language models. The details of the baseline algorithm are provided in the Appendix B.1.

**Metrics** Performance in puzzle and manipulation tasks is measured by the average Success Rate (SR), where a non-zero reward is provided only upon goal completion. For the SVG task, we employ a composite similarity metric aggregating DreamSim and DINO scores to assess generative quality. Detailed reward strategy and metrics definition are provided in Appendix B.2.

**Projector Architecture** We instantiate the projector $g_\psi$ as a lightweight 2-layer Multi-Layer Perceptron (MLP). To prevent representational collapse, it employs a bottleneck architecture, projecting the VLM's high-dimensional hidden state to a bottleneck layer (half the visual latent size) before mapping it to the visual feature space.

**Implementation Details** During training, we employ decoupled learning rates: $1 \times 10^{-6}$ for both the prediction loss

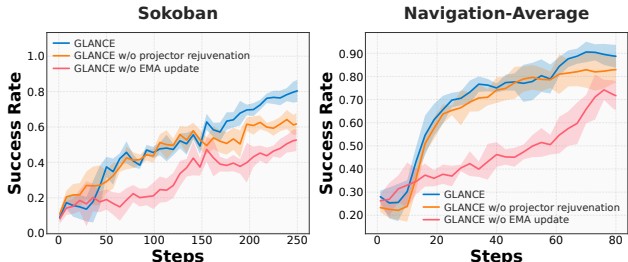

*Figure 3.* Ablation study on GLANCE components

optimization and the RL actor, while the critic is optimized with $1 \times 10^{-5}$; the global batch size is 128. To counter the non-stationarity and high variance of the training process, we employ the same reward normalization scheme as in (Schulman et al., 2017; Burda et al., 2019), dividing intrinsic rewards by the running standard deviation of their corresponding returns. To balance exploration and exploitation, we set the intrinsic reward scaling factor $\beta = 0.3$ for the Navigation task and $\beta = 0.1$ for all other environments. During inference and rollout, we set the VLM generation temperature to $0.7$ to balance diversity and precision. See Appendix B.8 for more implementation details.

### 4.2. Main Results

As shown in Tab. 1, our proposed GLANCE framework achieves consistently superior test-time performance across all five tasks compared to baselines relying solely on extrinsic rewards. In the dense extrinsic reward RL setting, GLANCE achieves an overall score of 0.86, a significant improvement over VAGEN-Full. This performance gap is particularly evident in challenging logic-heavy tasks like Sokoban, where our method improves the success rate from 0.79 to 0.85. These results validate our core hypothesis: aligning the agent's linguistic world model with visual reality provides a more robust grounding signal than passive exploitation of visited states.

Moreover, the effectiveness of our curiosity drive is most prominent in the Turn-level PPO setting, especially within the PrimitiveSkill environment. In PrimitiveSkill, where precise coordinate reasoning is required but task rewards are extremely sparse, purely exploitation-based agents often suffer from reasoning stagnation. GLANCE overcomes this by incentivizing the agent to actively visit and reason about unfamiliar object-hand interactions. This confirms that for VLM agents, curiosity is not merely an auxiliary bonus but a fundamental driver for discovering the complex transition dynamics necessary for long-horizon control.

### 4.3. Ablation Study

In this section, we conduct component-wise ablations to verify the efficacy of our architectural choices and the ro-

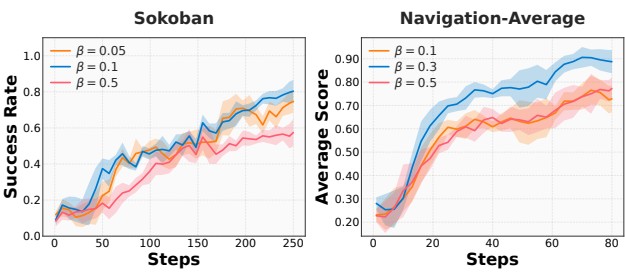

*Figure 4.* Sensitivity analysis of the exploration weight $\beta$.

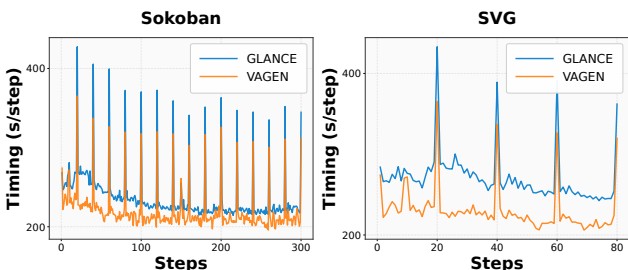

*Figure 5.* Comparison of per-step training time

bustness of the intrinsic reward. Unless otherwise specified, all results are averaged over 3 random seeds.

**Curriculum Exploration and Momentum Targets** We first conduct an ablation study on the Sokoban and Navigation tasks. As shown in Fig. 3, while the agent without adaptive curriculum rejuvenation enables decent initial learning, our approach achieves superior performance, particularly in the later stages of training. This indicates that the rejuvenation mechanism successfully prevents the intrinsic reward from diminishing too early. Furthermore, replacing the momentum target encoder with a direct online copy results in significant instability and divergence. This underscores that the slowly evolving teacher provides a critical, consistent anchor for grounding the agent's linguistic hypothesis.

*Table 2.* Ablation analysis on extrinsic rewards $r_t^{\text{reason}}$.

| Method | Sokoban$_{(\pm\text{std})}$ | FrozenLake | Navigation (Avg) |
|---|---|---|---|
| VAGEN-Full (w/ $r_t^{\text{reason}}$) | $\mathbf{0.76}_{\pm 0.070}$ | $\mathbf{0.70}_{\pm 0.055}$ | $\mathbf{0.76}_{\pm 0.076}$ |
| VAGEN-Full (w/o $r_t^{\text{reason}}$) | $0.62_{\pm 0.083}$ | $0.66_{\pm 0.048}$ | $0.71_{\pm 0.087}$ |
| GLANCE-Full (w/o $r_t^{\text{reason}}$) | $0.63_{\pm 0.076}$ | $0.68_{\pm 0.059}$ | $0.74_{\pm 0.071}$ |

**Intrinsic Curiosity vs. External Supervision** A central question is whether our self-supervised intrinsic reward can substitute for expensive dense rewards. Tab. 2 compares the impact of removing the dense reasoning reward $r_t^{\text{reason}}$ provided by the LLM-as-a-judge. As expected, removing $r_t^{\text{reason}}$ leads to a decrease in performance for both frameworks, confirming that external feedback is a strong training signal. However, in this challenging setting lacking reasoning supervision, GLANCE-Full consistently outperforms VAGEN-Full. This suggests that our curiosity-driven intrinsic reward provides a useful complementary signal to reasoning supervision. Even in the absence of an external judge, the drive to align "thinking" with "seeing" implicitly encourages the agent to maintain high-quality reasoning.

**Sensitivity to Exploration Weight ($\beta$)** Finally, we analyze the sensitivity of the exploration coefficient $\beta$. As shown in Fig. 4, we observe that the VLM performance peaks at moderate values, exhibiting a non-monotonic relationship. A moderate weight ($\beta \in [0.1, 0.3]$) strikes the op-

timal balance, providing sufficient epistemic drive without overshadowing the task reward. Conversely, an excessively high $\beta$ (0.5) distracts the agent, leading to noisy exploration, while a too-low $\beta$ fails to provide enough impetus to overcome exploration barriers. See Appendix B.8 for details on selecting the exploration weight $\beta$.

### 4.4. Further Analysis

**Computational Efficiency** A potential concern with incorporating self-supervised objectives is the added computational overhead. In Fig. 5, we compare the per-step training time of GLANCE against VAGEN-Full across two representative tasks. Although GLANCE introduces an additional forward pass for the momentum target network and a backward pass for the prediction loss, the computational overhead remains marginal. This efficiency is primarily attributed to our lightweight projector design and the fact that the target network requires no gradient computation. Given the significant performance gains, this trade-off is highly favorable for scaling VLM agents.

**Visualization of Curriculum Dynamics** To empirically validate the effectiveness of our *Curriculum Exploration* mechanism, we visualize the temporal correlation between the prediction loss and task success rate in Fig. 6. The results exhibit a distinctive sawtooth style pattern: upon each projector rejuvenation, the prediction loss intentionally spikes, reflecting a sudden rejuvenation of the intrinsic curiosity signal. Crucially, each loss spike is followed by a subsequent increase in the success rate, allowing the VLM agent to break through performance plateaus that frequently trap passive models. This provides evidence that our curriculum mechanism effectively sustains the epistemic drive, allowing the visual encoder to ground reasoning in increasingly fine-grained environment details without stagnating.

## 5. Related Work

**Curiosity-driven Exploration** Prediction error has been shown to be an effective indicator of novelty across different domains, including neuroscience (Oudeyer & Kaplan, 2007)

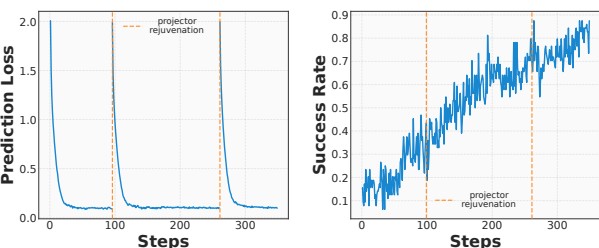

*Figure 6.* Visualization of the curriculum exploration dynamics.

and machine learning. In particular, it manifests itself in decision-making systems to encourage active exploration, for example, through competency maps (Thrun & Möller, 1991). The comparison between prediction and actual observation or reward leads to subjective surprise (Schmidhuber, 2010). This process interleaves with the world model, consistently treating prediction as an expectation of reality and, in turn, refining the world model itself (Schmidhuber, 1991). Similar ideas have been adopted in deep RL in combination with neural network–based modules (Stadie et al., 2015; Pathak et al., 2017; Burda et al., 2019; Guo et al., 2022), showcasing the capability to solve sparse-reward problems and to sufficiently explore complex environments that would otherwise be impossible for standard RL. Nonetheless, these methods are confined to either pixel- or proprioception-based domains. Adapting them to the semantic reasoning space of VLM agents, where uncertainty stems from linguistic-visual misalignment, remains an open frontier that our work addresses.

**World Models in Autonomous Agents.** World models, which learn to simulate environmental dynamics to support planning, have been central to model-based RL (Sutton, 1991; Ha & Schmidhuber, 2018). Traditional approaches like Dreamer (Hafner et al., 2019) and MuZero (Schrittwieser et al., 2020) learn the transition model in a latent space to perform lookahead search or generate synthetic experiences. With the advent of LLMs, a new paradigm has emerged where the pre-trained transformer itself acts as a world simulator. Methods such as RAP (Hao et al., 2023) and WebDreamer (Gu et al., 2024) leverage LLMs to predict action outcomes and score candidate plans in text-based or web environments. However, these approaches typically treat the world model as an external module or rely on frozen knowledge. Most recently, frameworks like VAGEN (Wang et al., 2025) have proposed internalizing world modeling directly into the VLM's policy via explicit Chain-of-Thought reasoning. While promising, these methods predominantly rely on passive supervision (e.g., LLM-as-a-judge) to refine the world model on visited states. In contrast, GLANCE integrates world modeling with curiosity-driven exploration, transforming the reasoning process from a passive inference

task into an active exploration process that proactively seeks to falsify and refine the agent's internal belief state.

## 6. Discussion and Limitations

In this work, we demonstrate that grounding linguistic reasoning in visual reality creates a robust epistemic drive. Below, we discuss critical design choices, current limitations, and future directions.

**Backbone Update and Language Drift** A central design choice in GLANCE is to freeze the LLM backbone while back-propagating gradients solely to the visual encoder and projector. While updating the LLM could allow the agent to refine its reasoning logic to better match visual dynamics, it introduces the risk of language drift. Furthermore, fine-tuning large-scale backbones via RL is computationally intensive. We hypothesize that a balanced approach like Low-Rank Adapters (LoRA) could modulate reasoning without catastrophic forgetting, representing a promising avenue for future efficiency-performance trade-offs.

**Projector Expressivity and Information Bottleneck** We currently employ a lightweight MLP projector to bridge the semantic-visual gap. This creates a tight information bottleneck that forces the visual encoder to capture high-level actionable semantics. However, this simple architecture might limit the granularity of the alignment. Future work could explore more sophisticated cross-modal mechanisms, such as Transformer-based projectors or cross-attention layers, to capture complex spatial-temporal correlations between the CoT and the future environment state. At the same time, GLANCE does not require both modalities to be highly accurate a priori. Instead, it explicitly leverages moderate cross-modal mismatch as a learning signal for grounding and exploration. Stable momentum targets and rejuvenation help make this signal more robust, although severe errors in either modality may still limit alignment quality.

## 7. Conclusion

We present GLANCE, a framework that enables VLM agents to explore by grounding linguistic predictions in visual reality. For this, we introduce a self-supervised alignment objective that turns the mismatch between what the agent predicts and what it sees into intrinsic rewards for grounded exploration. GLANCE achieves strong performance across sparse-reward agentic tasks and reduces reliance on costly dense reasoning supervision. We further show that *Adaptive Curriculum Rejuvenation* sustains exploration by revisiting uncertain states during long-horizon training. Overall, our results suggest that VLM agents should not only reason about the world, but actively test their internal hypotheses against visual evidence.

## Acknowledgements

This work was supported by fundings from the Hong Kong RGC General Research Fund (152228/23E, 162161/24E, 162116/25E, 162180/25E), National Natural Science Foundation of China (NSFC) Key Program (No. 62532005), Collaborative Research Fund (No. C1042-23GF, No. C5097-25G), NSFC/RGC Collaborative Research Scheme (Grant No. 62461160332 & CRS_HKUST602/24), Research Impact Fund (No. R5011-23F), Areas of Excellence Scheme (AoE/E-601/22-R), and the InnoHK (HKGAI).

## Impact Statement

This paper presents work whose goal is to advance the field of machine learning. There are many potential societal consequences of our work, none of which we feel must be specifically highlighted here.

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

# A. Algorithm

---

**Algorithm 1** GLANCE

---

**Input:** Learning rate for RL $\eta$ and visual-linguistic alignment $\kappa$, EMA decay $\alpha$, intrinsic reward scale $\beta$, curiosity drain parameters $\epsilon$ and $K$.
**Initialize:** VLM parameters $\boldsymbol{\theta} = \{\mathbf{v}, \boldsymbol{\ell}\}$, projector parameters $\boldsymbol{\psi}$, momentum encoder $\boldsymbol{\phi} \leftarrow \mathbf{v}$, and replay buffer $\mathcal{B} \leftarrow \emptyset$.
 1: **for** iteration $i = 1, 2, \ldots$ **do**
 2:     #1: Trajectory Rollout
 3:     **for** step $t = 1, \ldots, T$ **do**
 4:         Observe $o_t$ and generate response $a_t \sim \pi_{\boldsymbol{\theta}}(\cdot | o_t, h_t)$ with explicit prediction $s_{t+1}$.
 5:         Execute action $a_t$ in environment, receive $o_{t+1}$ and task reward $r_t^{\text{task}}$.
 6:         Extract online latent $h_{t+1}$ at `</prediction>` and target latent $y_{t+1} = \text{sg}(f_{\boldsymbol{\phi}}(o_{t+1}))$.
 7:         Compute prediction error $\mathcal{L}_{\text{explore}}(h_{t+1}, y_{t+1})$ and intrinsic reward $r_t^{\text{i}} = \beta \cdot \text{Normalize}(\mathcal{L}_{\text{explore}})$.
 8:         Store transition and latent pair in $\mathcal{B}$.
 9:     **end for**
10:     #2: Optimization
11:     **Step A: Joint Representation Learning**
12:         Freeze $\boldsymbol{\ell}$ to preserve linguistic reasoning.
13:         Sample latent pairs from $\mathcal{B}$, compute $\nabla\mathcal{L}_{\text{explore}}$ w.r.t. $\{\boldsymbol{\psi}, \mathbf{v}\}$.
14:         Update projector and visual encoder:
15:             $\boldsymbol{\psi} \leftarrow \boldsymbol{\psi} - \kappa\nabla_{\boldsymbol{\psi}}\mathcal{L}_{\text{explore}}$
16:             $\mathbf{v} \leftarrow \mathbf{v} - \kappa\nabla_{\mathbf{v}}\mathcal{L}_{\text{explore}}$
17:     **Step B: Policy and Critic Update**
18:         Compute Advantages $A_t$ using Bi-level GAE on $r_t^{\text{total}} = r_t^{\text{task}} + r_t^{\text{reason}} + r_t^{\text{format}} + r_t^{\text{i}}$.
19:         Unfreeze $\boldsymbol{\ell}$ and update full parameters $\boldsymbol{\theta}$ via PPO objective:
20:             $\boldsymbol{\theta} \leftarrow \boldsymbol{\theta} - \eta\nabla_{\boldsymbol{\theta}}\mathcal{J}_{\text{PPO}}(\boldsymbol{\theta})$
21:     **Step C: Momentum Encoder Update**
22:         $\boldsymbol{\phi} \leftarrow \alpha\boldsymbol{\phi} + (1 - \alpha)\mathbf{v}$
23:     #3: Rejuvenation (Optional)
24:     **if** `curiosity drain` **then**
25:         $\boldsymbol{\psi} \leftarrow \text{Randomly Initialize}(\boldsymbol{\psi})$
26:     **end if**
27:     Clear the buffer $\mathcal{B} \leftarrow \emptyset$.
28: **end for**

---

# B. Experimental Details.

### B.1. Baseline

To evaluate the robustness and plug-and-play nature of GLANCE, we benchmark against several RL baselines specifically designed for VLM agents or multi-turn reasoning. All trained baselines share the same `Qwen2.5-VL-3B` backbone and the internalized world-modeling action structure $a_t = \langle z_t, a_t^e \rangle$ defined in Section 2.2. The primary distinctions between these baselines lie in the composition of the extrinsic reward $r_t^e$ and the granularity of the credit assignment mechanism (i.e., advantage estimation). Table 3 summarizes these differences.

*Table 3.* **Key differences between RL baselines.** We categorize baselines by their extrinsic reward density and the level at which advantages are calculated and propagated.

| Baseline | Extrinsic Reward ($r_t^e$) | Advantage Estimation |
|---|---|---|
| VAGEN-Full (Wang et al., 2025) | Dense: $r_t^{\text{task}} + r_t^{\text{reason}} + r_t^{\text{format}}$ | Bi-Level GAE |
| VAGEN-Base (Wang et al., 2025) | Sparse: $r_t^{\text{task}} + r_t^{\text{format}}$ | Token-Level GAE |
| Turn-level PPO (Zhai et al., 2024) | Sparse: $r_t^{\text{task}} + r_t^{\text{format}}$ | Turn-Level GAE (Uniform) |

**VAGEN-Full (Wang et al., 2025)**   This baseline represents the upper bound of dense supervision. It utilizes a comprehensive reward signal where $r_t^{\text{reason}}$ is derived from an LLM-as-a-judge to provide direct feedback on the accuracy of state estimation and transition modeling tokens. To effectively bridge the gap between high-level reasoning turns and low-level token generation, it employs **Bi-Level GAE**. This mechanism first computes a turn-level advantage $A_t^{\text{turn}}$ and then uses it as a terminal target to compute a secondary, inner-GAE for individual reasoning tokens within $z_t$. This ensures that the credit for a successful turn is explicitly assigned to the reasoning steps that justified the action.

**VAGEN-Base (Wang et al., 2025)**   As a more challenging baseline for exploration, VAGEN-Base removes the dense reasoning supervision ($r_t^{\text{reason}} = 0$), relying only on task success and formatting adherence. Furthermore, it simplifies the credit assignment by using standard **Token-Level GAE**, where the sparse turn reward is assigned to the final token of the sequence. Advantages are then propagated backward through the temporal-difference (TD) error across all action tokens solely based on the value function's estimates. This setup tests the agent's ability to learn internal world models without explicit semantic guidance. This no-guidance setting also contrasts with a line of work (Yao et al., 2023; Liu et al., 2026) that explicitly structures or specifies agent cognition and behavior in VLM agents.

**Turn-level PPO with Masking (Zhai et al., 2024)**   This approach treats each turn as the atomic unit of optimization. While it uses the same sparse reward as VAGEN-Base, it simplifies the advantage estimation by computing a single scalar advantage for the entire turn ($A_t^{\text{turn}}$). This scalar is then assigned **uniformly** to every non-masked action token in the sequence. This coarse-grained assignment serves as a baseline to demonstrate the necessity of token-specific credit assignment for generating complex, multi-step reasoning chains.

**Standard RL Baselines**   In addition to the VLM-specific methods, we evaluate Vanilla PPO (Schulman et al., 2017) and GRPO (Shao et al., 2024). Vanilla PPO is implemented without observation token masking, highlighting the pitfalls of standard RL in long-context language modeling. GRPO is utilized as a reference for group-based relative policy optimization, which omits the critic network but often requires significantly larger sample sizes to resolve high trajectory diversity in visual environments.

### B.2. Reward and Evaluation Metrics

We follow the evaluation protocol of VAGEN (Wang et al., 2025) and use the same core metrics for comparability. We evaluate agents using trajectory-level metrics. Let $\tau$ denote a rollout trajectory, and let $\mathcal{D}$ denote the test set of trajectories. For evaluation, we define:

- $f(\tau) \in \{0, 1\}$: binary success indicator, equal to 1 if the task objective is achieved by the end of $\tau$ and 0 otherwise;

- $g(\tau) \in [0, 1]$: DreamSim similarity between the final rendered image and the target image;

- $h(\tau) \in [0, 1]$: DINO-based similarity between the final rendered image and the target image.

**Puzzle and manipulation environments.**   For Sokoban, FrozenLake, Navigation, and PrimitiveSkill, a trajectory is considered successful if it meets the environment-defined success condition. We report the average success rate:

$$\text{Success Rate} = \mathbb{E}_{\tau \sim \mathcal{D}}[f(\tau)]. \tag{10}$$

Consistent with these tasks, the environment reward is sparse and becomes non-zero only upon goal completion.

**SVG reconstruction.**   For SVG Reconstruction, we assess generation quality using perceptual similarity. We report the test-set averages of DreamSim and DINO similarities:

$$\text{DreamSim Score} = \mathbb{E}_{\tau \sim \mathcal{D}}[g(\tau)], \tag{11}$$
$$\text{DINO Score} = \mathbb{E}_{\tau \sim \mathcal{D}}[h(\tau)]. \tag{12}$$

When a single summary metric is needed, we additionally report Avg $=$ (DreamSim Score+DINO Score)$/2$. DreamSim (Fu et al., 2023) measures perceptual similarity via diffusion-model representations, while DINO (Caron et al., 2021) computes similarity in a self-supervised visual feature space; together they offer complementary views of reconstruction quality.

## B.3. Environment Settings

**Sokoban.** Tab. 4 lists the available actions, and Tab. 5 summarizes the environment configuration used in our experiments.

*Table 4.* Action space for the Sokoban environment.

| Name | Description |
| --- | --- |
| Up | Shift the agent by one grid cell upward. |
| Left | Shift the agent by one grid cell to the left. |
| Right | Shift the agent by one grid cell to the right. |
| Down | Shift the agent by one grid cell downward. |

*Table 5.* Hyperparameters for the Sokoban environment.

| Name | Value | Description |
| --- | --- | --- |
| dim_room | $(6, 6)$ | Grid size for the Sokoban room. |
| max_steps | 100 | Maximum number of steps per episode. |
| num_boxes | 1 | Number of boxes that must be pushed onto targets. |
| min_actions_to_succeed | 5 | Minimum action count required for a successful episode. |
| max_actions_per_step | 3 | Maximum number of actions allowed per turn. |
| max_turns | 3 | Maximum number of interaction turns per episode. |

**FrozenLake.** We use a four-direction grid controller (Tab. 6); key environment settings are shown in Tab. 7.

*Table 6.* Action space for the FrozenLake environment.

| Name | Description |
| --- | --- |
| Up | Move one cell toward the upper neighbor. |
| Left | Move one cell toward the left neighbor. |
| Right | Move one cell toward the right neighbor. |
| Down | Move one cell toward the lower neighbor. |

*Table 7.* Hyperparameters for the FrozenLake environment.

| Name | Value | Description |
| --- | --- | --- |
| desc | None | Map layout (None indicates a randomly generated map). |
| is_slippery | False | Whether transitions include slipping dynamics. |
| size | 4 | Side length of the square grid. |
| max_actions_per_step | 3 | Maximum number of actions allowed per turn. |
| min_actions_to_succeed | 5 | Minimum action count required to count as success. |
| max_turns | 3 | Maximum number of interaction turns per episode. |

**Navigation.** Tab. 8 enumerates movement and camera controls, while Tab. 9 reports the rendering and control settings.

*Table 8.* Action space for the Navigation environment.

| Name | Description |
|------|-------------|
| MoveAhead | Translate forward by one movement step. |
| MoveBack | Translate backward by one movement step. |
| MoveRight | Strafe right by one movement step. |
| MoveLeft | Strafe left by one movement step. |
| RotateRight | Yaw right by $90°$. |
| RotateLeft | Yaw left by $90°$. |
| LookUp | Pitch the camera upward by $30°$. |
| LookDown | Pitch the camera downward by $30°$. |

*Table 9.* Hyperparameters for the Navigation environment.

| Name | Value | Description |
|------|-------|-------------|
| resolution | 255 | Rendered image resolution. |
| down_sample_ratio | 1.0 | Down-sampling ratio for observations. |
| fov | 100 | Camera field-of-view in degrees. |
| multiview | False | Whether multiple camera views are enabled. |
| max_actions_per_step | 5 | Maximum number of actions allowed per turn. |
| success_threshold | 1.5 | Distance threshold for success. |
| step_length | 0.5 | Translation distance per movement action. |
| max_turns | 4 | Maximum number of interaction turns per episode. |

**PrimitiveSkill.** The agent issues parameterized manipulation commands (Tab. 10). We also list the per-turn action budget and horizon in Tab. 11.

*Table 10.* Action space for the PrimitiveSkill environment.

| Name | Description |
|------|-------------|
| pick(x, y, z) | Grasp an object at position $(x, y, z)$ in the workspace. |
| place(x, y, z) | Place the grasped object at position $(x, y, z)$. |
| push(x1, y1, z1, x2, y2, z2) | Push an object from $(x1, y1, z1)$ to $(x2, y2, z2)$. |

*Table 11.* Hyperparameters for the PrimitiveSkill environment.

| Name | Value | Description |
|------|-------|-------------|
| max_actions_per_step | 2 | Maximum number of actions allowed per turn. |
| max_turns | 3 | Maximum number of interaction turns per episode. |

**SVG Reconstruction.** The policy outputs free-form SVG markup as its action (Tab. 12); dataset and horizon details are provided in Tab. 13.

*Table 12.* Action space for the SVG Reconstruction environment.

| Name | Description |
|------|-------------|
| SVG Code | A text action specifying SVG markup. |

*Table 13.* Hyperparameters for the SVG Reconstruction environment.

| Name | Value | Description |
|---|---|---|
| dataset_name | starvector/svg-icons-simple | Dataset used to sample target SVG examples. |
| max_turns | 2 | Maximum number of interaction turns per episode. |

## B.4. Reward Assignment

**Sokoban.** Tab. 14 details the reward components used in Sokoban.

*Table 14.* Reward structure for the Sokoban environment.

| Reward Type | Value | Description |
|---|---|---|
| Success reward | 10 | Granted when all boxes are placed on target locations. |
| Failure penalty | -0.1 | Applied each step until completion. |
| Box placement reward | 1 | Added for each box pushed onto a target. |
| Format reward | 0.5 | Per-turn reward encouraging structured visual state reasoning. |
| Grounding reward weight | 0.5 | Weight applied to the task-state reward. |
| World modeling reward weight | 0.5 | Weight applied to the task-transition reward. |

**FrozenLake.** The reward decomposition for FrozenLake is summarized in Tab. 15.

*Table 15.* Reward structure for the FrozenLake environment.

| Reward Type | Value | Description |
|---|---|---|
| Success reward | 10 | Granted when the agent reaches the goal position. |
| Failure penalty | -0.1 | Applied each step until completion. |
| Format reward | 0.5 | Per-turn reward encouraging structured visual state reasoning. |
| Grounding reward weight | 0.5 | Weight applied to the task-state reward. |
| World modeling reward weight | 0.5 | Weight applied to the task-transition reward. |

**Navigation.** The reward components for Navigation are listed in Tab. 16.

*Table 16.* Reward structure for the Navigation environment.

| Reward Type | Value | Description |
|---|---|---|
| Success reward | 10 | Granted when the agent reaches the goal location. |
| Failure penalty | -0.1 | Applied each step until completion. |
| Format reward | 0.5 | Per-turn reward encouraging structured visual state reasoning. |
| Grounding reward weight | 0.5 | Weight applied to the task-state reward. |
| World modeling reward weight | 0.5 | Weight applied to the task-transition reward. |

**PrimitiveSkill.** Tab. 17 reports the reward design for PrimitiveSkill.

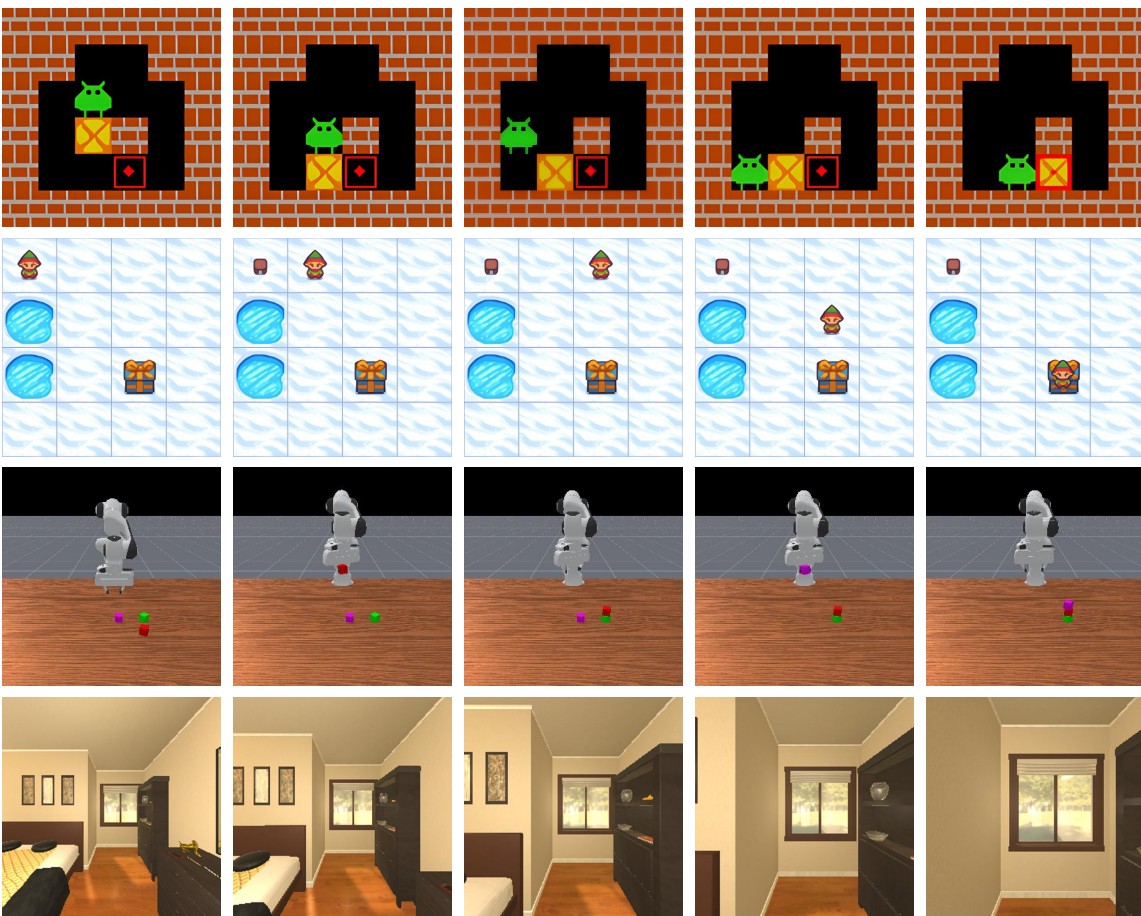

*Figure 7.* Examples of visual states from four environments used in our study

*Table 17.* Reward structure for the PrimitiveSkill environment.

| Reward Type | Value | Description |
| --- | --- | --- |
| Success reward | 10 | Granted when the manipulation task is completed. |
| Failure penalty | -0.1 | Applied each step until completion. |
| Stage-based reward | $(\text{stage} + 1) \times 2$ | Granted upon reaching key subgoals (stage is the highest completed stage). |
| Format reward | 0.5 | Per-turn reward encouraging structured visual state reasoning. |
| Grounding reward weight | 0.5 | Weight applied to the task-state reward. |
| World modeling reward weight | 0.5 | Weight applied to the task-transition reward. |

**SVG Reconstruction.** The SVG reward is based on visual similarity; Tab. 18 lists the full specification.

*Table 18.* Reward structure for the SVG Reconstruction environment.

| Reward Type | Value | Description |
|---|---|---|
| Image similarity | Variable | Weighted DreamSim (Fu et al., 2023) and DINO (Caron et al., 2021) similarity between the generated and target images. |
| Format reward | 0.5 | Per-turn reward encouraging structured visual state reasoning. |
| Grounding reward weight | 0.5 | Weight applied to the task-state reward. |
| World modeling reward weight | 0.5 | Weight applied to the task-transition reward. |
| DreamSim weight | 5.0 | Scale factor applied to DreamSim similarity. |
| Dino weight | 0.0001 | We only use DreamSim score for reward. |

## B.5. Additional Multi-Seed Results

In this section, we report additional results in the dense extrinsic reward setting, comparing VAGEN-Full and GLANCE-Full. All values are reported as mean ± std over 3 independent seeds. Standard Sokoban, FrozenLake, Navigation, and PrimitiveSkill correspond to the same settings as Table 1. To further examine robustness beyond the standard setting, we additionally include *Noisy Sokoban* with Gaussian observation noise and *Slippery FrozenLake* with stochastic transitions. We also report results on EB-ALFRED (Yang et al., 2025), where the *Base* subset evaluates basic task-solving ability and the *Long Horizon* subset contains tasks requiring extended action sequences, typically more than 15 steps.

*Table 19.* Additional multi-seed results in the dense extrinsic reward setting. We report mean ± std over 3 independent seeds for VAGEN-Full and GLANCE-Full. Standard Sokoban/FrozenLake/Navigation/PrimitiveSkill correspond to the same settings as Table 1. Noisy Sokoban uses Gaussian observation noise $\mathcal{N}(0, \sigma^2 I)$ with $\sigma = 0.02$, and Slippery FrozenLake uses stochastic transitions with $P(\text{slip}) \in \{0.02, 0.04\}$. EB-ALFRED reports the Base and Long Horizon subsets, and each subset consists of 50 test instances. Bold indicates the better trained model in each column.

| Model/Method | Sokoban | | FrozenLake | | | Navigation | PrimitiveSkill | EB-ALFRED | |
|---|---|---|---|---|---|---|---|---|---|
| | Standard | Noisy | Standard | Slippery (.02) | Slippery (.04) | Avg. | Avg. | Base | Long Horizon |
| Dense Extrinsic Rewards RL with World Model Reasoning for Visual States (Backbone: QWEN2.5-VL-3B; 3 seeds) | | | | | | | | | |
| VAGEN-Full | $0.76_{\pm 0.070}$ | $0.59_{\pm 0.048}$ | $0.70_{\pm 0.055}$ | $0.63_{\pm 0.047}$ | $0.54_{\pm 0.039}$ | $0.76_{\pm 0.076}$ | $0.90_{\pm 0.075}$ | $0.57_{\pm 0.117}$ | $0.54_{\pm 0.035}$ |
| GLANCE-Full | $0.83_{\pm 0.062}$ | $0.67_{\pm 0.035}$ | $0.75_{\pm 0.058}$ | $0.70_{\pm 0.041}$ | $0.60_{\pm 0.052}$ | $0.82_{\pm 0.050}$ | $0.92_{\pm 0.046}$ | $0.61_{\pm 0.103}$ | $0.59_{\pm 0.042}$ |

**Results.** Overall, GLANCE-Full consistently outperforms VAGEN-Full across the standard 3-seed settings, suggesting that the gains in Table 1 are not due to single-seed variance. Under stochastic transitions and visual perturbations, both methods degrade, indicating that such settings remain a genuine challenge for curiosity-driven exploration. However, GLANCE degrades less and maintains a clear advantage, suggesting the robustness of our intrinsic signal settings. Finally, GLANCE also improves over VAGEN on both the Base and Long Horizon subsets of EB-ALFRED, indicating that the method remains beneficial beyond short-horizon grid-style tasks and extends to more complex long-horizon embodied settings.

## B.6. Case Study

In this section, we show some cases from our four environments (Fig. 7).

## B.7. Prompt Collection

This section summarizes the prompt set used in our framework. We directly adopt the prompt templates from VAGEN (Wang et al., 2025) as our default prompts. For clarity and conciseness, we only present the prompts corresponding to Sokoban, Frozenlake, and Navigation in this section.

**Sokoban Training Prompt for World Model Strategy**

```
You are a Sokoban solver.
Sokoban Quick Guide
Goal: Push all boxes onto targets.
Symbols (If image is provided there are no symbols):
# Wall | _ Floor | O Target | X Box | P You | * Box on Target | S You on Target
```

```
Rules:
1. Push boxes (can't pull).
2. Avoid walls.
Actions you can take: Left, Down, Right, Up.
You can take up to 3 action(s) at a time, separated by ,.
You should first give the description of your observation, then your reasoning, then
    predict the next state, and finally your answer.
Your response should be in the format of:
<think><observation>...</observation><reasoning>...</reasoning>
<prediction>...</prediction></think><answer>...</answer>
e.g. <think><observation>The box is below the player and the target is below the box</
    observation><reasoning>I need to go down then push the box down to the target</
    reasoning><prediction>The player will be above the box, the target and box will be
    at the same place.</prediction></think><answer>Down,Down</answer>
[Initial Observation]:
<image>
Decide your next action(s).
You can take up to 3 action(s) at a time, separated by ,.
You should first give the description of your observation, then your reasoning, then
    predict the next state, and finally your answer.
Your response should be in the format of:
<think><observation>...</observation><reasoning>...</reasoning>
<prediction>...</prediction></think><answer>...</answer>
<think><observation>The player is at the bottom of the screen, and there is a box to
    the right of the player. The target is to the left of the box.</observation><
    reasoning>The player needs to push the box to the target to complete the goal.</
    reasoning><prediction>The player will push the box to the target, moving up, down,
    and to the left.</prediction></think><answer>Up, Down, Left</answer>
```

**FrozenLake Training Prompt for World Model Strategy**

```
You are a FrozenLake solver.
FrozenLake Quick Guide
Goal: Reach the goal (G).
Symbols (If image is provided there are no symbols):
_ Frozen | O Hole | G Goal | P Player | X Player fell into hole | * Player on goal
Rules:
1. Avoid falling into holes.
2. Frozen tiles are slippery, you may move perpendicular to your intended direction.
Actions you can take: Left, Down, Right, Up.

You can take up to 3 action(s) at a time, separated by ,.
You should first describe the observation, then your reasoning, then predict the next
    state, and finally your answer.
Your response should be in the format of:
<think><observation>...</observation><reasoning>...</reasoning>
<prediction>...</prediction></think><answer>...</answer>
e.g. <think><observation>The player is on the above the target</observation><reasoning
    >I should go down then left to reach the target</reasoning><prediction>The player
    will reach the target</prediction></think><answer>Down,Left</answer>
[Initial Observation]:
<image>
Decide your next action(s).

You can take up to 3 action(s) at a time, separated by ,.
You should first describe the observation, then your reasoning, then predict the next
    state, and finally your answer.
Your response should be in the format of:
<think><observation>...</observation><reasoning>...</reasoning>
<prediction>...</prediction></think><answer>...</answer>
<think><observation>The player is on the right side of the grid.</observation><
    reasoning>The player is on the right side of the grid, which is indicated by the
```

```
        position on the grid.</reasoning><prediction>The player will move to the left or
        down.</prediction></think><answer>Left, Left, Down</answer>
```

---

**Navigation Training Prompt for World Model Strategy**

```
You are a home robot and perform navigation tasks according to instructions.
Actions you can take: moveahead, moveback, moveright, moveleft, rotateright,
    rotateleft, lookup, lookdown.
moveahead: Move forward by some distance
moveback: Move backward by some distance
moveright: Move rightward by some distance
moveleft: Move leftward by some distance
rotateright: Rotate to the right by 90 degrees
rotateleft: Rotate to the left by 90 degrees
lookup: Tilt the camera upward by 30 degrees
lookdown: Tilt the camera downward by 30 degrees
Rewards:
Format correct: +0.5
Achieve the human instruction: +10.0
The instruction will be provided with each observation. Look at the image carefully
    and navigate to complete the instruction.
Hints:
1. You can take multiple actions at a time, in most cases, if you find the target
    object is far away from you, you can call moveahead, moveleft and move right
    multiple times.
2. If you find yourself seems to be stuck, you can lookdown to see if there's any
    object above or below you, you can also rotate to see if there's any object behind
    you.
Example:
Round 1:
image_1
<think><observation>There is a garbage can in the upper left corner of the image, next
    to the kitchen sink. To move there, we can go forward-left, but since there's a
    kitchen counter directly ahead, we should go left first.</observation><reasoning>
    Following the strategy, I can go by first moving leftward.</reasoning><prediction>I
    will be infront of the garbage</prediction></think>
<answer>moveleft, moveleft</answer>
Round 2:
Env_feedback: Last action is executed successfully.
image_2
<think><observation>From the secene, I see that by moving leftward, we are getting
    closer to the garbage can. Now, the garbage can is in front of me, slightly to the
    left. And there's a large area ahead of us.</observation><reasoning>Following the
    strategy, I can go by first moving forward then moving leftward.</reasoning><
    prediction>I will be closer to the garbage</prediction></think>
<answer>moveahead, moveahead,moveahead,moveleft</answer>
Round 3:
Env_feedback: Last action is executed successfully.
image_3
<think><observation>From the image we can see the garbage can is very close to us,
    still to our front-left. Moving leftward might be blocked but i can see that there
    is still space in front of me to get closer to the garbage can.</observation><
    reasoning>Following the strategy, we can take about two steps forward then one step
     left to reach the garbage can.</reasoning><prediction>I will reach the garbage</
    prediction></think>
<answer>moveahead, moveahead,moveleft</answer>
Round 4:
Env_feedback: Success
You can take up to 5 action(s) at a time, separated by ','.
You should first give your thought process with the your observation, reasoning, and
    prediction of next state, then your answer.
```

```
Both the observation and prediction should describe what you see or expect to see in
    the environment.
Your response should be in the format of:
<think><observation>...</observation><reasoning>...</reasoning>
<prediction>...</prediction></think><answer>...</answer>
e.g. <think><observation>I am at the entrance of a bedroom. There is a bed to the left,
    a desk with a lamp on the right, and a closet straight ahead. The target object, a
    book, appears to be on the desk.</observation><reasoning>I need to move toward the
    desk to reach the book. I'll turn right and move forward.</reasoning><prediction>I
    am now standing in front of the desk. The desk has a lamp, a computer, and several
    books on it. The target book is within reach on the right side of the desk.</
    prediction></think><answer>rotateright,moveahead,moveahead</answer>
[Initial Observation]:
<image>
Human Instruction: I am looking for a luminous device on my desk to read books at
    night. Can you navigate to that object and stay close?
Decide your next action(s).
You can take up to 5 action(s) at a time, separated by ','.
You should first give your thought process with the your observation, reasoning, and
    prediction of next state, then your answer.
Both the observation and prediction should describe what you see or expect to see in
    the environment.
Your response should be in the format of:
<think><observation>...</observation><reasoning>...</reasoning>
<prediction>...</prediction></think><answer>...</answer>
<think><observation>There is a luminous device on the desk to the right of the chair.
    To reach it, I should move forward and then turn right.</observation><reasoning>
    Following the strategy, I can go by first moving forward then turning right to
    reach the luminous device.</reasoning><prediction>I will be close to the luminous
    device</prediction></think><answer>moveahead, moveahead, moveright</answer>
```

We use an LLM-as-a-Judge to evaluate the agent's reasoning by checking whether the agent's descriptions and predictions match the ground-truth state information. We use task-specific prompts for both and StateEstimation (current-state grounding) and TransitionModeling (next-state prediction). Each prompt asks the judge to review the evidence and then output a YES/NO decision.

**Sokoban STATEESTIMATION Evaluation**

```
Evaluate whether the description accurately captures the key position relationships in
     the Sokoban game state.
Answer YES if the directional relationships are correct, or NO if they contain
    directional errors.

# Context
You are evaluating whether the description correctly identifies the directional
    relationships between:
1. The player and the box(es)
2. The box(es) and the target(s)

The description doesn't need to be perfectly precise or mention every detail – it just
     needs to have the correct directional relationships (Up, Down, Left, Right).

Example:
Groundtruth Current State Information: ['box0 is at the same row and to the left of
    the player', 'target0 is above and on the left side of the player', 'target0 is
    above and on the left side of box0']
State Description: The player is below the box and the target is below the box.
– <think>The state description contains spatial relationship information, do further
    analysis. According to the ground truth data, box0 is at the same row and to the
    left of the player, target0 is above and on the left side of the player, target0 is
     above and on the left side of box0. The description states 'The player is below
    the box and the target is below the box.' The player is actually at the same row as
```

```
      the box (not below), and the target is actually above the box (not below). Both
      directional relationships are incorrectly identified.</think><answer>NO</answer>

Example:
Groundtruth Current State Information: ['box0 is above and on the right side of the
      player', 'target0 is above and at the same column as the player', 'target0 is above
       and on the left side of box0']
State Description: The box is above the player and the target is to the left of the
      box
- <think>The state description contains spatial relationship information, do further
      analysis. According to the ground truth data, box0 is above and on the right side
      of the player, target0 is above and at the same column as the player, target0 is
      above and on the left side of box0. The description states 'The box is above the
      player and the target is to the left of the box.' It correctly identifies that the
      box is above the player (box0 is above and on the right side of the player). It
      also correctly identifies that the target is to the left of the box (target0 is
      above and on the left side of box0). Both key directional relationships are
      accurately described.</think><answer>YES</answer>

# Groundtruth Current State Information:
{state_information_dict}

# State Description:
{natural_language_description}

Think step by step:
  1. Relative Relationship Requirements:
    - Must describe at least one relationships BETWEEN entities (player-box, player-
    target, box-target)
    - Absolute positions like "player is on the left side" are insufficient
    - Need relational descriptions like "player is left of target"

  2. Essential Relationships to Check
    - Player-Target relationship (highest priority)
    - Player-Box relationship
    - Box-Target relationship

  3. Equivalent Expression Recognition
    - "box is above player" = "player is below box"
    - "target is left of box" = "box is right of target"
    - Must recognize these as identical spatial relationships. Absolute position is
    not allowed

Your answer should be in the format of <think>...</think><answer>YES</answer> or <
    think>...</think><answer>NO</answer>.
```

## Sokoban TRANSITIONMODELING Evaluation

```
Evaluate whether the prediction correctly anticipates the key position relationships
    that will exist in the next Sokoban state.
Answer YES if the predicted directional relationships are correct, or NO if they
    contain directional errors.

# Context
You are evaluating whether the prediction correctly identifies the directional
    relationships that will exist after the move:
1. The future position of the player relative to the box(es)
2. The future position of the box(es) relative to the target(s)

# Important: The Prediction Comes First
Remember: The Next State Prediction is made BEFORE the Groundtruth Next State exists.
    Your task is to check if the prediction correctly anticipated what actually
```

```
    happened.
If the box and target are at same position, this prediciton is seen as success
    immediately (YES)

Example:
Groundtruth Next State Information: ['box0 is above and on the right side of the
    player', 'target0 is above and on the left side of the player', 'target0 is above
    and on the left side of box0']
Next State Prediction: The player will be to the left of the box, and the box will be
    to the right of the target.
- <think>The prediction state contains spatial relationship between player and target,
     do further analysis. According to the ground truth data, box0 is above and on the
    right side of the player, target0 is above and on the left side of the player,
    target0 is above and on the left side of box0. The description states 'The player
    will be to the left of the box, and the box will be to the right of the target.' It
     correctly identifies that the player is to the left of the box (since box0 is on
    the right side of the player). It also correctly identifies that the box is to the
    right of the target (since target0 is on the left side of box0). Therefore, this
    description correctly identifies the key directional relationships.</think><answer>
    YES</answer>

# Groundtruth Next State Information:
{state_information_dict}

# Next State Prediction:
{natural_language_description}

Think step by step:
  1. Relative Relationship Requirements:
    - Must describe at least one relationships BETWEEN entities (player-box, player-
    target, box-target)
    - Absolute positions like "player is on the left side" are insufficient
    - Need relational descriptions like "player is left of target"

  2. Essential Relationships to Check
    - Player-Target relationship (highest priority)
    - Player-Box relationship
    - Box-Target relationship

  3. Equivalent Expression Recognition
    - "box is above player" = "player is below box"
    - "target is left of box" = "box is right of target"
    - Must recognize these as identical spatial relationships. Absolute position is
    not allowed

Your answer should be in the format of <think>...</think><answer>YES</answer> or <
    think>...</think><answer>NO</answer>.
```

**FrozenLake STATEESTIMATION Evaluation**

```
Evaluate whether the description accurately captures the key position relationships in
     the FrozenLake game state.
Answer YES if the directional relationships are correct, or NO if there are errors.

# Context
You are evaluating whether the description correctly identifies:

1. The directional relationship between the player and the goal (MUST Have)
2. The directional relationship between the player and the hole (if present)

The description doesn't need to be perfectly precise - it just needs to have the
    correct directional relationships between the player and target (Up, Down, Left,
```

```
        Right), and between the player and hole if applicable.

# Groundtruth Current State Information:
{state_information_dict}

# State Description:
{natural_language_description}

Think step by step:
1. Player relationship with Goal
   - Goal (Target) MUST include in state description, without target the description is
      automatically wrong (NO)
   - If there is no direction between player and goal, like "player is right to the
     target", the description is automatically wrong (NO)
   - This takes highest priority over all other considerations

2. Equivalent Expression Recognition
   - "goal is above player" = "player is below goal"
   - "target is left of box" = "box is right of target"
   - Must recognize these as identical spatial relationships. Absolute position is not
     allowed

3. Simple Judgment Rule
   - If player at goal -> YES
   - If direction aligns with needed movement -> YES
   - Otherwise -> NO

Your answer should be in the format of <think>...</think><answer>YES</answer> or <
    think>...</think><answer>NO</answer>.
```

---

**FrozenLake TRANSITIONMODELING Evaluation**

```
Evaluate whether the prediction correctly anticipates the key aspects of the next
    FrozenLake state.
Answer YES if the prediction accounts for directional relationships and potential
    holes, or NO if it contains errors.

# Context
You are evaluating whether the prediction correctly identifies:
1. The position relationship between the player and the goal after the prediction

# Important: The Prediction Comes First
Remember: The Next State Prediction is made BEFORE the Groundtruth Next State exists.
    Your task is to check if the prediction correctly anticipated what actually
    happened.

The prediction doesn't need to perfectly describe every aspect of the next state - it
    just needs to correctly anticipate the directional relationships (Up, Down, Left,
    Right) or address any dangers from holes.

# Groundtruth Next State Information:
{state_information_dict}

# Next State Prediction:
{natural_language_description}

Think step by step:
1. Player relationship with Goal
   - If player is already at the goal position, the prediction is automatically correct
      (YES)
   - Goal (Target) MUST include in prediction state, without target the prediction is
     automatically wrong (NO)
```

```
    - If there is no direction between player and goal, like "player is right to the
      target", the prediction is automatically wrong (NO)
    - This takes highest priority over all other considerations

2. Directional Correctness
    - Evaluate if the predicted movement direction aligns with the relative position
      between player and goal
    - For example, if player is left of goal, moving right is correct
    - **CRITICAL: Recognize equivalent expressions of the same spatial relationship**
      * "player is above target" = "target is below player"
      * "player is left of target" = "target is right of player"
      * These are the SAME relationship expressed from different perspectives

3. Simple Judgment Rule
    - If player at goal -> YES
    - If direction aligns with needed movement -> YES
    - Otherwise -> NO

Your answer should be in the format of <think>...</think><answer>YES</answer> or <
    think>...</think><answer>NO</answer>.
```

## Navigation STATEESTIMATION Evaluation

```
Evaluate whether the description effectively communicates the spatial relationship
    between the agent and target object, even if the exact directional terms differ.
Answer YES if the overall spatial understanding is correct, or NO if it fundamentally
    misunderstands the spatial layout.

# Context
You are evaluating whether the description effectively conveys where the target object
     is located relative to the agent. The exact directional terminology (left, right,
    ahead, etc.) may differ between the state information and the description, but the
    important factor is whether the description would lead to correct navigation.

# Groundtruth Current State Information:
{state_information_dict}

# State Description:
{natural_language_description}

Think step by step:
1. Check if the description contains spatial relationship between agent and target
    object
    - If no spatial relationship is mentioned, answer NO
2. If spatial relationship exists, check if the predicted direction is consistent with
     the target direction
    - "ahead/forward" = "ahead"
    - "left" = "left"
    - "right" = "right"
    - Combined directions like "forward-left", "forward-right" are acceptable if they
      include the correct primary direction
3. The prediction is correct if it mentions moving toward the target in a direction
    that reasonably aligns with the groundtruth direction

Your answer should be in the format of <think>...</think><answer>YES</answer> or <
    think>...</think><answer>NO</answer>.
```

## Navigation TRANSITIONMODELING Evaluation

```
Evaluate whether the prediction effectively anticipates how the agent would navigate
    toward the target object, even if the exact directional terms differ.
Answer YES if the overall navigation plan is reasonable, or NO if it misunderstands or
```

```
    did not mention the spatial layout.

# Context
You are evaluating whether the prediction effectively anticipates how the agent would
    move to reach the target object. The exact directional terminology (left, right,
    ahead, etc.) may differ between the state information and the prediction, but the
    important factor is whether the prediction would lead to successful navigation.

# Important: The Prediction Comes First
Remember: The Next State Prediction is made BEFORE the Groundtruth Next State exists.
    Your task is to check if the prediction correctly anticipated what actually
    happened.

# Groundtruth Next State Information:
{state_information_dict}

# Next State Prediction:
{natural_language_description}

Think step by step:
1. First, check if the prediction explicitly uses EXACT directional terms that appear
    in the groundtruth state: "ahead", "left", "right", "up", "down".
  - Terms like "move towards", "closer to", "near", "approaching", "in front of", "by",
    "at" DO NOT qualify
  - "Will be on the left/right/ahead" or "Will move left/right/forward" DO qualify
  - If no exact directional match to groundtruth is present, conclude with NO
    immediately
2. If explicit direction words exist, verify that they EXACTLY match the target object'
    s direction in the groundtruth:
  - If target is "ahead", prediction must specify "ahead", "forward", "slightly left",
    OR "slightly right" (special case: we allow slightly left/right for ahead targets)
  - If target is "right", prediction must specify "right"
  - If target is "left", prediction must specify "left"
3. Even if the prediction mentions intermediate objects correctly, it MUST explicitly
    state the correct final direction to the target object
4. The prediction cannot substitute object references for directions (saying "move to
    X" instead of "move right")
5. Remember that the prediction was made BEFORE the groundtruth state was determined

Your answer should be in the format of <think>...</think><answer>YES</answer> or <
    think>...</think><answer>NO</answer>.
```

## B.8. Implementation Details

**Rejuvenation Settings**   We set the convergence threshold $\epsilon = 0.1$, for consecutive duration steps, we use $K = 30$ for Sokoban and FrozenLake and $K = 20$ for the other tasks.

**Exploration Weight Selection.**   We select the intrinsic reward scaling factor $\beta$ via a line search over $\{0.05, 0.1, 0.3, 0.5, 1.0\}$. Based on this search, we set $\beta = 0.3$ for Navigation and $\beta = 0.1$ for all other environments. These values consistently provide a moderate exploration drive without overwhelming the task reward.

**Projector Architecture.**   We use a lightweight two-layer MLP projector to map the VLM hidden representation to the target visual feature space. Specifically, we take the final-layer hidden state associated with the `</prediction>` token as the prediction representation, denoted by $x = h_{t+1} \in \mathbb{R}^{d_{\text{vlm}}}$, and produce a visual space prediction $\hat{y}_{t+1} \in \mathbb{R}^{d_{\text{vis}}}$.

The projector is

$$\text{Proj}(x) = W_2\, \sigma\left(\text{LN}(W_1 x)\right),$$

implemented as `Linear` $\to$ `LayerNorm` $\to$ `ReLU` $\to$ `Linear`. In all experiments, we set $d_{\text{vlm}} = 2048$ and $d_{\text{vis}} = 2048$.

