# OpenReview forum: "What You Think is What You See: Driving Exploration in VLM Agents via Visual-Linguistic Curiosity"
_ICML.cc/2026/Conference — ICML 2026 spotlight_

### Official Review · Reviewer_YSrD · 2026-03-12

**Soundness:** 3
**Presentation:** 2
**Significance:** 3
**Originality:** 2
**Overall Recommendation:** 4
**Confidence:** 2

**Summary:**

This paper introduces GLANCE, a curiosity-driven framework for VLM agents that unifies world modeling and exploration by aligning an agent’s linguistic prediction of future state (from its Chain-of-Thought reasoning) with the visual representation of the next observation encoded by a momentum target network. The discrepancy between the projected linguistic latent and the target visual latent serves both as a self-supervised alignment loss (to shape the vision encoder) and as an intrinsic reward (to drive exploration). To prevent early collapse of curiosity (“curiosity drain”), the authors propose a curriculum-style projector rejuvenation that periodically reinitializes the language-to-vision projector when the prediction loss stagnates. Experiments across grid puzzles, 3D navigation, manipulation, and SVG reconstruction report consistent improvements over exploitation-based VLM-RL baselines and demonstrate the role of momentum targets and rejuvenation.

**Compliance With Llm Reviewing Policy:**

Affirmed.

**Key Questions For Authors:**

- How sensitive is performance to the choice of the linguistic latent (last prediction token) and tokenization format? Have you tried pooling multiple prediction tokens or using attention-weighted aggregates?
- Can you report multi-seed means and standard deviations for the main results, plus learning curves with confidence intervals? What is the typical sample budget per task?
- How does GLANCE compare to a carefully tuned BYOL-Explore (visual-only) baseline and to ICM/RND adapted to your VLM setting? This is critical to isolate the contribution of cross-modal alignment over generic curiosity.
- How robust is the intrinsic signal under stochasticity (e.g., slippery FrozenLake, action noise, or visual distractors)? Do rejuvenation resets exacerbate or mitigate noisy-TV behavior?
- Can you clarify the timing spikes in Fig. 5 and provide hardware details? Are spikes correlated with rejuvenation events or target-network synchronization?
- Did you experiment with adaptive β (e.g., EIPO-style dual updates) or learning-progress–based rewards for the cross-modal loss? If so, how did they compare?
- What are the failure cases qualitatively? Do agents ever “hallucinate” predictions that are easily misaligned to boost intrinsic reward? How do you detect or prevent such behavior?
- Could small, regularized updates to the LLM (e.g., LoRA) further improve alignment without language drift? Any preliminary findings?

**Limitations:**

yes

**Strengths And Weaknesses:**

Technical novelty and innovation
- Introduces a cross-modal curiosity signal that explicitly aligns “what the agent thinks” (linguistic next-state latent) with “what it sees” (visual latent), going beyond visual-only prediction errors (e.g., ICM, BYOL-Explore).
Uses a momentum target encoder to stabilize the cross-modal regression target, adapting self-supervised vision learning ideas to VLM agent settings.
- Proposes a simple but effective “Curriculum Exploration” via projector rejuvenation to mitigate premature convergence of the intrinsic signal.
Experimental rigor and validation
- Evaluates across heterogeneous domains (discrete puzzles, embodied 3D navigation, parameterized robotic manipulation, and generative SVG) with a unified 3B VLM backbone.
- Includes ablations on key design elements (EMA target, projector rejuvenation) and sensitivity to intrinsic reward weight β.
Compares under both dense- and sparse-reward RL settings and shows that the intrinsic signal can partially compensate for the absence of dense reasoning supervision.

---

> ### Author Rebuttal · Authors · 2026-03-31
>
> We sincerely thank Reviewer YSrD for the constructive feedback. We are encouraged that the reviewer finds our contributions novel and the empirical evidence strong.
>
> The main concerns involve the stochastic robustness of our method, the clarity of experimental details, and the need for deeper analysis of several results and design choices. We address these concerns below with clarifications and new experimental results.
>
> > **Robustness under Stochastic Setting (Q4)**
>
> We thank the reviewer for raising this important question about the stochastic robustness of our method.
>
> To address this concern, we conducted experiments on **slippery FrozenLake** ($P(\text{slip}) = 0.02, 0.04$) and **Sokoban with visual distractors** ($\mathcal{N}(0, \sigma^2 I), \sigma = 0.02$). Results are shown below (2 seeds, 240 steps).
>
> |Method|Slippery FL (0.02)|Slippery FL (0.04)|Noisy Sokoban|
> |-|-:|-:|-:|
> |VAGEN|0.652 ± 0.035|0.559 ± 0.012|0.617 ± 0.039|
> |Ours|0.723 ± 0.020|0.619 ± 0.036|0.684 ± 0.020|
> |Ours w/o rejuvenation|0.705 ± 0.025|0.610 ± 0.040|—|
>
> Our method shows smaller degradation and outperforms VAGEN under stochastic settings. A possible reason is that our intrinsic signal is driven by the *linguistic prediction* and *next-step visual representation*, which may be more robust than traditional curiosity loss.
>
> We also observe that **rejuvenation** better sustains useful exploration under mild stochasticity. However, it is not a dedicated defense design against noisy-TV distractions.
> > **The Contribution of Cross-modal Alignment (Q3)**
>
> This is a critical question about the design of our framework. We compared our method with RND (w/o visual target) and BYOL-Explore (w/o language).
>
> - RND: same VLM visual encoder, followed by a predictor head and a random target head.
> - BYOL-Explore: observation representation concatenated with executable action embedding, followed by a 2-layer MLP predictor and the same projector as GLANCE.
>
> |Method|Sokoban|
> |-|-:|
> |RND|0.66|
> |BYOL|0.68|
> |Ours w/o rejuvenation|0.82|
>
> The results show the effectiveness of our cross-modal alignment mechanism.
> > **Multi-seed statistics and training budget (Q2)**
>
> We appreciate the reviewer’s request for multi-seed statistics. In the main paper, to maintain consistency and enable direct comparison with VAGEN and related baselines, we reported the same single-seed setting as prior work. Here, we report results across 3 seeds with mean and std:
>
> |Task|VAGEN|Ours|
> |-|-|-|
> |Sokoban|0.760 ± 0.070|0.830 ± 0.062|
> |Navigation|0.757 ± 0.076|0.817 ± 0.050|
> |PrimitiveSkill|0.900 ± 0.075|0.920 ± 0.046|
>
> These results show that our method performs better on average across seeds.
>
> Regarding the **sample budget**, we follow the standard VLM-agent setting from prior work. The global training batch size is **128**. The maximum number of turns ranges from **2–4** across tasks. Training runs for **300 iterations**. We will include multi-seed statistics and learning curves with confidence intervals in the revised version.
> > **Investigation on LoRA (Q8)**
>
> This is an interesting question. We conducted preliminary experiments with **LoRA** (r=32, α=16):
>
> |Method|Sokoban|Prediction Success|
> |-|-:|-:|
> |Ours (LoRA)|0.796|0.83|
> |Ours|0.823|0.91|
>
> These results suggest that LoRA may not improve performance, possibly because the policy loss and prediction loss share the same LoRA parameters, leading to gradient competition.
> > **Sensitivity to the linguistic latent choice (Q1)**
>
> We thank the reviewer for this insightful question. In a decoder-style Transformer, We think the last prediction token can naturally aggregate the history information. We compared this design with the mean pooling:
>
> |Method|Sokoban|
> |-|-:|
> |Mean|0.80|
> |Last|0.82|
>
> The results suggest that last-token representation performs slightly better in our current setting.
> > **Clarification of timing spikes(Q5)**
>
> We thank the reviewer for pointing this out. The timing spikes in *Fig. 5* are caused by periodic validation runs, triggered every *20 training steps*.
> > **Hallucination risk and the choice of β (Q6 & Q7)**
>
> Those are thoughtful questions. Since VLMs themselves may hallucinate, it is possible that our prediction error also reflects such behavior. However, in our framework, the extrinsic reward monitors prediction quality via both task completion and world-model reasoning accuracy. Moreover, the LLM backbone is **frozen** during prediction-loss optimization, so the agent cannot directly exploit language-model parameter updates to systematically generate hallucinated predictions for higher intrinsic reward.
>
> For the choice of β, we use the *absolute cross-modal prediction error* as the intrinsic reward, which is simple and effective in our settings. At the same time, we agree that rewarding *improvement in prediction* rather than raw prediction error may reduce attraction to unlearnable noisy transitions. We see this as a promising direction for future research.

---

### Official Review · Reviewer_4GPU · 2026-03-13

**Soundness:** 2
**Presentation:** 3
**Significance:** 2
**Originality:** 3
**Overall Recommendation:** 5
**Confidence:** 4

**Summary:**

This paper introduces GLANCE, a plug-and-play module that brings curiosity-driven exploration to VLM-based agents. Unlike existing curiosity-based exploration methods that operate entirely in visual latent spaces, VLM agents encounter a modality mismatch where observations are pixels, but world modeling is conducted in language space. GLANCE addresses this by predicting next state in language space, projecting the resulting representation into visual space, and computing the discrepancy against the actual next observation encoded by a target  visual encoder. This discrepancy serves as both representation learning objective and intrinsic curiosity reward. The paper also identifies a curiosity drain issue specific to VLM agents, and proposes periodic re-initialization of the projection layer to sustain exploration.

**Compliance With Llm Reviewing Policy:**

Affirmed.

**Final Justification:**

The rebuttal has fully addressed my concerns. I have updated my score to positive.

**Key Questions For Authors:**

The paper motivates GLANCE under a POMDP formulation, but Sokoban and FrozenLake provide fully observable grid views. Could the authors clarify what "lack of a global view" means in the Experimental Setup section (L267-269)?

**Limitations:**

While the paper discusses architectural limitations, it lacks discussion about scalability to more complex environments. Table 2 shows that the proposed curiosity signal provides marginal benefits on Sokoban, a logically hard task.

**Strengths And Weaknesses:**

### Strengths
- This paper identifies a gap in applying curiosity-driven exploration to VLM agents, where the observation and world modeling modalities are mismatched.
- The proposed method is well-motivated and intuitive.
- GLANCE can be applied on diverse VLM agent baselines without requiring modifications, and shows consistent improvement over baselines.

### Weaknesses
- While GLANCE improves over baselines across tasks, the gaps are often small. No error bars or confidence intervals are provided, making it difficult to assess whether these differences are statistically significant.
- All tasks in the experiments have small state spaces and short horizons. It is unclear whether GLANCE would be effective in exploration-hard environments such as Montezuma's Revenge, which is a representative benchmark for exploration methods.
- This paper claims that GLANCE achieves great performance without expensive dense reasoning reward ($r^{\text{reason}}_t$), but Table 2 shows a large gap between VAGEN-Full and GLANCE-Full in Sokoban. The gap between GLANCE-Full and VAGEN-Full without the reasoning reward is marginal. This raises concerns about whether the proposed method can provide meaningful learning signals in more complex settings.

---

> ### Author Rebuttal · Authors · 2026-03-31
>
> We sincerely thank Reviewer 4GPU for the detailed and constructive feedback. We are encouraged that the reviewer finds our method well-motivated and intuitive, and recognizes that GLANCE is plug-and-play across diverse VLM-agent baselines while showing consistent improvements.
>
> The reviewer's main concerns are related to the lack of uncertainty quantification, limited evaluation on short-horizon tasks, and whether GLANCE provides sufficiently meaningful learning signals w/o dense reasoning reward. We address these concerns below with clarifications and new experimental results.
>
> > **Multi-seed statistics (W1)**
>
> We appreciate the reviewer’s request for multi-seed reporting. In the main paper, to maintain consistency and enable direct comparison with VAGEN and related baselines, we followed the same single-seed setting as prior work. Here we additionally report mean ± std over 3 seeds, together with 95% confidence intervals:
>
> |Task|VAGEN|Ours|
> |-|-|-|
> |Sokoban|0.760 ± 0.070, CI: [0.68, 0.84]|0.830 ± 0.062, CI: [0.76, 0.90]|
> |Navigation|0.757 ± 0.076, CI: [0.67, 0.84]|0.817 ± 0.050, CI: [0.76, 0.87]|
> |PrimitiveSkill|0.900 ± 0.075, CI: [0.82, 0.98]|0.920 ± 0.046, CI: [0.87, 0.97]|
>
> These results confirm that our method performs better on average across seeds.
>
> Due to computational constraints, we cannot currently run multiple seeds for all tasks, baselines, and ablations during rebuttal. Each full run requires about **50 H200 GPU hours**. We will include additional multi-seed results and learning curves with CI in the revised version
> > **Beyond small-state, short-horizon tasks (W2)**
>
> We thank reviewer for raising this important question. Here we argue that the 3D embodied tasks Navigation and PrimitiveSkill do not have small state spaces; in contrast, their observations are high-dimensional, resulting in a large state space. To address the concern regarding the long-horizon capability, we conduct experiments on **EB-ALFRED**, an embodied benchmark with **7 household task types**, requiring long-horizon planning (more than 15 steps) with rich interactions involving different objects. For rebuttal efficiency, we focus on the *Base* and *Long Horizon* subsets:
>
> |Method|Base|Long Horizon|
> |-|-:|-:|
> |VAGEN|0.64|0.55|
> |Ours|0.67|0.61|
>
> Additionally, we evaluate on a challenging **stochastic** variant of FrozenLake to test robustness under sparse rewards and transition uncertainty:
>
> |Method|Slippery FL(0.02)|Slippery FL(0.04)|
> |-|-:|-:|
> |VAGEN|0.652 ± 0.035|0.559 ± 0.012|
> |Ours|0.723 ± 0.020|0.619 ± 0.036|
>
> Together, these results support the scalability and robustness of our method. We agree that **Montezuma’s Revenge** is a representative hard-exploration benchmark. It is not directly compatible with our current VLM-agent setting and at least needs thousands of GPU hours. We leave direct evaluation on such Atari-style settings to future work.
> > **Role of intrinsic reward (W3)**
>
> We thank the reviewer for this insightful question. Our goal is **not** to claim that GLANCE fully replaces dense reasoning reward, but rather to show that **its intrinsic reward remains effective even without $r_t^{reason}$**, and it still outperforms VAGEN in this setting.
>
> GLANCE uses the discrepancy between **linguistic prediction** and the next-step visual representation as its intrinsic signal, improving cross-modal grounding and exploration. Since this signal depends on the quality of the linguistic prediction, better **world-model reasoning quality** can also lead to better alignment quality. In this sense, $r_t^{reason}$ and GLANCE are complementary rather than redundant: $r_t^{reason}$ directly improves reasoning quality, while GLANCE turns prediction-reality mismatch into an intrinsic exploration signal. We will revise the claim in the line 363-376 to make this point more precise.
>
> > **POMDP Formulation (Q1)**
>
> We thank the reviewer for raising this key question. Our POMDP formulation is intended as a **general framework** across all tasks, especially to unify grid-world tasks and embodied 3D tasks under a common sequential decision-making perspective. For Navigation and other embodied tasks, the agent is indeed partially observable.
>
> For **Sokoban** and **FrozenLake**, we agree that the environments are not partially observable in the classical RL sense when a full grid view is provided. Our phrase **lack of a global view** was therefore imprecise. What we intended to emphasize is that the agent is trained under the same history-based policy framework. We will revise this Experiment Setup section to make the distinction clear.
>
> On top of that, even when the full view might be available, the mapping from observation to state is nontrivial, as are unknown transitions and rewards. In such cases, the problem can be effectively modeled as a POMDP [1-2].
>
> **Reference:**
>
> [1] *Comprehensive Benchmarking Multi-modal Large Language Models for Vision-driven Embodied Agents*, ICML 2025.
>
> [2] *Bayes-adaptive POMDPS*, NIPS 2007.

---

> > ### Author Rebuttal · Reviewer_4GPU · 2026-04-04
> >
> > Thank you for the detailed rebuttal. Most of my concerns have been resolved. I have updated my score.

---

> > > ### Author Response · Authors · 2026-04-04
> > >
> > > Dear Reviewer 4GPU,
> > >
> > > We sincerely thank you for recognizing the value of our work and for confirming that our rebuttal effectively addressed most of your concerns. The issues you raised regarding our method's statistical reliability, scalability, and broader evaluation beyond short-horizon tasks are all very important. We are pleased that our additional experiments and analyses helped clarify these points. We will incorporate these new results and revised formulation into the paper to fully reflect your constructive feedback.
> > >
> > > Best,
> > >
> > > Authors

---

### Official Review · Reviewer_ZSvW · 2026-03-13

**Soundness:** 4
**Presentation:** 3
**Significance:** 3
**Originality:** 3
**Overall Recommendation:** 5
**Confidence:** 4

**Summary:**

This paper introduces GLANCE, a unified framework that aims to bridge reasoning and exploration by grounding the agent’s linguistic world model with a target visual encoder. The idea is to align the predictions produced by the VLM with the visual observations from the environment. The discrepancy between linguistic predictions and visual reality is used as an intrinsic curiosity signal within an RL framework to encourage exploration that improves alignment between modalities. The method is evaluated across five agentic tasks, with the proposed approach outperforming several baselines in most scenarios.

**Compliance With Llm Reviewing Policy:**

Affirmed.

**Final Justification:**

The authors addressed my questions, but there is no grounds to raise the score above what has already been given.

**Key Questions For Authors:**

- Curiosity-driven exploration methods often suffer from white-noise attraction, where agents repeatedly visit unpredictable but uninformative states. Could this phenomenon occur with the proposed modality-discrepancy reward? If so, how could it be mitigated?
- Since the intrinsic reward is computed during task execution alongside extrinsic rewards, could the framework also be applied as a pretraining stage based solely on intrinsic curiosity, before learning specific tasks?
- How was the exploration weight selected for each environment? Is it the same?
- How many random seeds or independent runs were used for each experiment?
- The paper claims that the computational overhead is marginal, yet the reported runtime appears to significantly increase compared to VAGEN. Could the authors clarify this point?
- Could the curiosity-driven alignment mechanism be used to improve the language backbone, potentially serving as a pretraining objective for future models?

**Limitations:**

- The paper would benefit from a clearer discussion of potential failure modes of curiosity-driven exploration, particularly in environments containing stochastic or visually unpredictable states.
- The computational overhead introduced by the framework may limit its applicability in large-scale or real-time settings.
- The approach relies on the quality of both linguistic predictions and visual representations, which may restrict performance if either modality is inaccurate.
- The experiments do not explicitly analyze how the proposed intrinsic reward behaves in long-horizon or highly stochastic environments, where curiosity-driven exploration may become unstable.

**Strengths And Weaknesses:**

Strengths

The idea of using cross-modal discrepancy between linguistic prediction and visual observations as a curiosity signal is not entirely new, but is very well-used in the proposed framework. In fact, modality alignment is key for successful behaviour and using curiosity to address mismatches and enhance learning was very insigthful.  Another strong aspect of the work is the comprehensive experimental evaluation. The paper includes ablation studies that showcase the role of different components of the proposed framework. Also, the attempt to integrate reasoning capabilities with embodied learning is an important challenge in building more capable agentic systems. I beleive this work is very favaroble in this direction. The framework also appears to have a modular design, compatible with existing RL pipelines, what can drive further integrations.

--------------
Weaknesses

Curiosity-driven exploration methods are known to sometimes focus on stochastic or noisy states that are difficult to predict but not necessarily useful for solving the task. It is unclear whether the proposed modality-discrepancy signal may lead to similar behaviors, where the agent is attracted to visually unpredictable situations rather than meaningful exploration that improves task performance. The paper states that the computational overhead remains marginal, but the reported runtime suggests that the method may approximately double the computational cost compared to the baseline (VAGEN) in some settings. A clearer discussion of the trade-off between improved exploration and computational cost would strengthen the evaluation. Also, the choice of the exploration weight appears to significantly affect performance across environments. However, the paper does not clearly explain how this hyperparameter is selected per environment. Finally, although PPO is used as the underlying RL algorithm, the paper does not clearly state how many training runs or random seeds were used for each experiment. Since PPO can be sensitive to initialization and stochasticity, reporting results across multiple seeds is important for ensuring fair comparisons.

---

> ### Author Rebuttal · Authors · 2026-03-31
>
> > **Results on stochastic and long-horizon environments (Q1, L1, L4)**
>
> We thank the reviewer for raising this important question regarding the stochastic nature of our method. However, we argue that unpredictability does not imply uninformativity; in fact, some environments have optimal rewards that are inherently more stochastic. Moreover, efficient exploration is a necessary condition for optimality. Without it, the agent fails to solve the task regardless of whether it is stochastic. This implies that our method remains effective for stochastic tasks. Nonetheless, higher aleatoric uncertainty requires more time to learn, and if only a bounded budget is available, this could degrade performance.
>
> To validate this, we conducted experiments on slippery FrozenLake ($P(\text{slip}) = 0.02, 0.04$) and Sokoban with visual distractors ($\mathcal{N}(0, \sigma^2 I), \sigma = 0.02$). Results are shown below (2 seeds, 200 steps).
>
> |Method|Slippery FL(0.02)|Slippery FL(0.04)|Noisy Sokoban|
> | - | -: | -: | -: |
> |VAGEN|0.652 ± 0.035|0.559 ± 0.012|0.617 ± 0.039|
> |Ours|0.723 ± 0.020|0.619 ± 0.036|0.684 ± 0.020|
>
> As expected, there is some performance degradation, but our method still outperforms VAGEN, demonstrating the benefit of exploration in stochastic environments. In addition, our intrinsic signal is driven not only by the *next-step visual representation* but also by the *linguistic prediction*, whose pretrained knowledge may help enhance the robustness of our method.
>
> Next, we conduct experiments on **EB-ALFRED**, an embodied benchmark requiring long-horizon planning (more than 15 steps) with rich interactions involving different objects:
>
> |Method|Base|Long Horizon|
> |-|-:|-:|
> |VAGEN|0.64|0.55|
> |Ours|0.67|0.61|
>
> The results demonstrate that our method is better at handling long-horizon tasks.
> > **The Choice of the Exploration Weight (Q3)**
>
> Thank you for this valuable question. We conducted a line search over the range $\\{0.05, 0.1, 0.3, 0.5, 1.0\\}$ to select the optimal value of $\beta$. We found that $\beta = 0.3$ performs best for Navigation, while $\beta = 0.1$ works well for the remaining environments. We will clarify this selection procedure in the revised version.
> > **Multi-seed Results (Q4)**
>
> We appreciate the reviewer’s request for multi-seed reporting. In the main paper, to maintain consistency and enable direct comparison with VAGEN and related baselines, we followed the same single-seed setting as prior work. Here we report mean ± std over 3 seeds:
>
> |Task|VAGEN|Ours|
> |-|-|-|
> |Sokoban|0.760 ± 0.070|0.830 ± 0.062|
> |Navigation|0.757 ± 0.076|0.817 ± 0.050|
> |PrimitiveSkill|0.900 ± 0.075|0.920 ± 0.046|
>
> These results confirm that our method performs better on average across seeds. we will include additional multi-seed results and learning curves with confidence intervals in the revised version.
> > **Clarification of Computational Overhead (Q5, L2)**
>
> This is a crucial question. Here we checked our code details and find the reason: The target encoder runs on *CPU* due to a Ray process scheduling constraint, leading to over 71% of GLANCE-specific overhead in current version. Now, we have migrated this module to GPU，and it brings the total overhead well below the 2× estimate (e.g., from 17 GPU hours to 10 GPU hours on svg).
> > **Pretraining based on Intrinsic Curiosity (Q2, Q6)**
>
> This is an excellent question. It is definitely applicable to the pretraining stage, either as an objective or a framework. This can be viewed as a form of "active learning" [1], where the agent focuses more on samples that are less predictable or more uncertain. By maximizing the prediction error, the agent can identify such samples and allocate more computational resources to them—either by assigning higher weights or increasing their sampling probability [2]. In this way, the more uncertain samples receive greater attention, improving overall prediction quality [3].
> > **Reliance on modality quality (L3)**
>
> We thank the reviewer for this important aspect. However, GLANCE is not designed under the assumption that both modalities are already highly accurate. Rather, its key idea is to leverage the discrepancy in cross-modal alignment. Consequently, moderate mismatches are not purely a weakness; they actually drive grounding and exploration.
>
> Moreover, GLANCE employs a **momentum target encoder**, which provides stable visual targets, and **rejuvenation**, which helps handle known unknowns. Therefore, GLANCE is designed to convert moderate *cross-modal mismatch into a useful training signal* while reducing fragility through stable targets and selective gradient routing. We will include this discussion in the revised version.
>
> **Reference**:
>
> [1] Active Learning Literature Survey, Technical Report 2009
>
> [2] Prioritized Experience Replay, ICLR 2016
>
> [3] Deep Bayesian Active Learning with Image Data, ICML 2017

---

> > ### Author Rebuttal · Reviewer_ZSvW · 2026-04-03
> >
> > I thank the authors for their rebuttal. They have addressed all the additional questions I raised. Given the current manuscript and the rebuttals, I believe my scores are appropriate.

---

> > > ### Author Response · Authors · 2026-04-03
> > >
> > > Dear Reviewer ZSvW,
> > >
> > > We sincerely thank you for recognizing the value of our work and for confirming that our rebuttal effectively addressed most of your concerns. The issues you raised regarding method applicability, computational overhead, scalability, and robustness are all very important. We are pleased that our additional experiments and analyses helped clarify these points. We will incorporate these new results and discussions into the paper to ensure that your constructive feedback is fully reflected.
> > >
> > > Best,
> > > Authors

---

### Official Review · Reviewer_5FMV · 2026-03-19

[review text omitted: it was posted to a different submission]

---

> ### Author Rebuttal · Authors · 2026-03-31
>
> We sincerely thank the reviewer for their time and effort to evaluate our submission.
> However, we would like to respectfully ask whether this review may have been inadvertently attached to the wrong submission.
>
> Our paper studies **GLANCE**, a curiosity-driven framework for VLM agents that uses cross-modal discrepancy between linguistic prediction and visual observation as an intrinsic exploration signal. In contrast, the current review discusses a rather different setting, including an **Virtual Model Learning**,  **an independently trained world model**, **PUCT search**, and the **DeepPHY benchmark**, which do not appear in our manuscript.
>
> We therefore wonder whether the comments may correspond to another submission. If so, we would be grateful if the reviewer could kindly revisit both the comments and the corresponding scores based on the correct manuscript.
>
> We truly appreciate the reviewer’s understanding.

---

### Decision · Program_Chairs · 2026-04-30

**Decision:**

Accept (spotlight)

**Comment:**

This paper introduces GLANCE, a plug-and-play, curiosity-driven framework for VLM-based agents that unifies world modeling and exploration. The initial ratings for this paper were split, with two weak rejects, one weak accept, and one accept. One of the reviewers who gave a weak reject raised concerns such as the lack of statistical significance analysis, the effectiveness in exploration-hard environments, and the large performance gap depending on reward selection. However, these concerns were well addressed in the rebuttal, leading this reviewer to raise the score to accept. The comments from the other reviewer who gave a weak reject were, in fact, substantially mismatched with the content of this paper, containing references to topics such as Virtual Model Learning, an independently trained world model, PUCT search, and the DeepPHY benchmark, none of which appear in this work. Therefore, it is possible that this reviewer mistakenly submitted comments intended for a different paper. As this reviewer did not submit a rebuttal acknowledgement (despite a reminder from the AC), their evaluation is not considered in the final decision. Following the evaluations of the remaining three reviewers, the AC recommends acceptance of this paper.